# Hydrophobic pore gates regulate ion permeation in polycystic kidney disease 2 and 2L1 channels

Wang Zheng[1,2], Xiaoyong Yang[3], Ruikun Hu[4], Ruiqi Cai[2], Laura Hofmann[5], Zhifei Wang[6], Qiaolin Hu[2], Xiong Liu[2], David Bulkley[7], Yong Yu[6], Jingfeng Tang[1], Veit Flockerzi[5], Ying Cao[4], Erhu Cao[3] & Xing-Zhen Chen[1,2]

*PKD2* and *PKD1* genes are mutated in human autosomal dominant polycystic kidney disease. PKD2 can form either a homomeric cation channel or a heteromeric complex with the PKD1 receptor, presumed to respond to ligand(s) and/or mechanical stimuli. Here, we identify a two-residue hydrophobic gate in PKD2L1, and a single-residue hydrophobic gate in PKD2. We find that a PKD2 gain-of-function gate mutant effectively rescues PKD2 knockdown-induced phenotypes in embryonic zebrafish. The structure of a PKD2 activating mutant F604P by cryo-electron microscopy reveals a π- to α-helix transition within the pore-lining helix S6 that leads to repositioning of the gate residue and channel activation. Overall the results identify hydrophobic gates and a gating mechanism of PKD2 and PKD2L1.

[1] National "111" Center for Cellular Regulation and Molecular Pharmaceutics, Hubei University of Technology, Wuhan, Hubei 430068, China. [2] Department of Physiology, Membrane Protein Disease Research Group, Faculty of Medicine and Dentistry, University of Alberta, Edmonton, AB T6G 2H7, Canada. [3] Department of Biochemistry, University of Utah School of Medicine, Salt Lake City, UT 84112, USA. [4] School of Life Sciences and Technology, Tongji University, Shanghai 200092, China. [5] Experimentelle und Klinische Pharmakologie und Toxikologie, Universität des Saarlandes, Homburg 66421, Germany. [6] Department of Biological Sciences, St. John's University, Queens, NY 11439, USA. [7] Keck Advanced Microscopy Laboratory and Department of Biochemistry and Biophysics, University of California, San Francisco, San Francisco, CA 94143, USA. These authors contributed equally: Wang Zheng, Xiaoyong Yang. Correspondence and requests for materials should be addressed to J.T. (email: jingfeng9930@163.com) or to E.C. (email: erhu.cao@biochem.utah.edu) or to X.-Z.C. (email: xzchen@ualberta.ca)

Autosomal dominant polycystic kidney disease (ADPKD) is a prominent human monogenic disease and affects over 12.5 million people worldwide[1]. It is characterized by the formation of fluid-filled renal cysts, leading to an increase in the total kidney volume and a decline in the renal function. Approximately 50% of ADPKD patients ultimately develop end stage renal disease and require dialysis or renal transplantation. ADPKD is, however, a systemic disease and its extra-renal manifestations include hepatic and pancreatic cysts, cerebral aneurysms, cardiovascular abnormalities and hypertension[2].

ADPKD is caused by mutations in either the *PKD1* or *PKD2* gene encoding PKD1 and PKD2, respectively[3]. PKD1, also called polycystin-1, is a receptor-like protein with 11 transmembrane segments, a large extracellular N-terminus predicted to bind unknown ligand(s) and a short intracellular C-terminus[3]. PKD2, also called polycystin-2 or TRPP2, is a cation channel of the transient receptor potential polycystin (TRPP) subfamily of the TRP superfamily[4]. Recent evidences showed that PKD2 has high permeability to $Na^+$ and $K^+$, but low permeability to $Ca^{2+}$ [5,6]. PKD2 shares an overall architecture with other TRP channels, i.e., six transmembrane segments (S1–S6), a pore loop between S5 and S6, and intracellularly localized N- and C-termini[7–9]. PKD1 and PKD2 have been shown to form a complex[10–12], which is hypothesized to sense fluid flow on the membrane of primary cilia. However, this hypothesis remains debatable[13–17]. Clinically identified pathogenic mutations in either *PKD1* or *PKD2* were thought to result in abnormal PKD1/PKD2 channel function and dysregulated cilium signaling, ultimately initiating cyst formation. There is no cure for ADPKD to date. The outcomes of recent clinical trials have been mostly negative, with the exception of the marginally effective Tolvaptan being approved in some countries to treat ADPKD[18]. Therefore, understanding how the PKD1/PKD2 complex functions is crucial for elucidating the pathogenesis of ADPKD and developing therapeutic strategies.

PKD2 has been shown to assemble as either homotetrameric ion channels[7–9] or heterotetrameric complexes with PKD1[11,12]. The heterologously expressed PKD1/PKD2 complex in HEK293 or CHO cells was recently shown to be activated by WNT molecules that bind to the PKD1 N-terminus[19]. Whether WNTs serve as physiological ligands for the ciliary PKD1/PKD2 complex has yet to be determined. Intracellular $Ca^{2+}$ was reported to increase the open probability of homotetrameric PKD2 channels at low concentrations while inhibiting channel opening at elevated levels[20]. By directly measuring currents in primary cilia of renal collecting duct epithelial cells of conditional PKD1 or PKD2 knockout mice, PKD2 was recently shown to be an essential $K^+$ and $Na^+$ permeable ion channel in primary cilia independent of PKD1[6]. Using screening mutagenesis, we identified an activating mutant F604P that gave rise to robust cation currents when expressed in *Xenopus* oocytes, allowing us to further investigate PKD2 channel function[5].

PKD2L1 (also called polycystin-2L1 or TRPP3) is a homologue of PKD2 (54% amino acid identity), but is not involved in ADPKD. PKD2L1 was initially reported to be a $Ca^{2+}$-activated non-selective cation channel in *Xenopus* oocytes[21]. When heterologously expressed in HEK293 cells, PKD2L1 also mediated measurable non-selective cation currents[22,23]. PKD2L1 associates with PKD1L1, a homologue of PKD1, in primary cilia and acts as a $Ca^{2+}$ channel regulating ciliary $Ca^{2+}$ concentration and hedgehog signaling[24,25]. PKD2L1, in complex with PKD1L3 or alone, is activated by protons in an off-response manner, i.e., activation occurs only after low extracellular pH is removed[26,27].

The molecular gating mechanisms of PKD2 and PKD2L1 channels have remained poorly understood. We recently identified and characterized hydrophobic gate (i.e., lower gate or activation gate) residues in the distal part of the pore-lining S6 helix

of several TRP channels[28]. A hydrophobic gate residue allows the pore to close by forming a so-called hydrophobic barrier, preventing ion permeation even when the physical pore size is bigger than that of the ion[29]. Whether PKD2 and PKD2L1 adopt a similar hydrophobic gating mechanism is unknown. We[9] and two other groups[7,8] recently resolved PKD2 structures and revealed two constriction points in the distal portion of S6 formed by residues L677 and N681. However, it remains unclear whether both L677 and N681 or just one of them function as a gate in PKD2. Among the four PKD2 structures reported to date, three of them seem to be in a non-conducting state while the fourth one exhibits a wider pore opening. Whether the structure with the wider opening represents an activated state remains unclear. An extracellular tetragonal opening for polycystins (TOP) domain between the S1 and S2 helices contributes to channel assembly and likely participates in recognition of unidentified ligand(s)[7–9], but whether and how this domain is implicated in PKD2 channel activation has yet to be determined. Therefore, a PKD2 structure in an activated state is needed to provide insights into the PKD2 gating mechanism and associated conformational changes.

In this study, we reveal a PKD2 gating mechanism using a combination of functional and structural analyses. We first identify and analyze a series of gain-of-function (GOF) and loss-of-function (LOF) mutants of the hydrophobic residues in the distal part of the pore-lining S6 helix in PKD2L1 and PKD2, revealing that the two residues (L557-A558) and L677 residue alone form a hydrophobic gate in PKD2L1 and PKD2, respectively. We then show that PKD2 GOF gate mutants are able to rescue PKD2 knockdown-induced disease phenotypes in zebrafish. Lastly, we obtain a 3.5 Å structure of the PKD2 activating mutant F604P by single-particle cryo-electron microscopy (EM) and reveal that a transition from π- to α-helix in the middle of S6 leads to expansion of the activation gate via a combined twisting and splaying motion of the gate residues.

## Results

**Identification of hydrophobic gate residues in PKD2L1.** According to the hydrophobic gate theory[29], replacement of a hydrophobic gate residue with a hydrophilic residue result in hydrophobic energy barrier collapse and an elevated hydration rate of the pore, thus giving rise to a constitutively open channel. To examine whether PKD2L1 adopts a hydrophobic gate, we replaced each of the eight hydrophobic residues (L552-I560) in the distal region of the pore-lining S6 helix (Fig. 1a) with hydrophilic asparagine (N) and assessed the channel activity of the resulting point mutants in *Xenopus* oocytes by two-electrode voltage clamp electrophysiology. We found that compared with wild-type (WT) PKD2L1, two mutants, L557N (the PKD2L1 L557 residue corresponds to the PKD2 L677 which forms a pore constriction[9]) and A558N, exhibited dramatic $9.0 \pm 1.4$-fold and $11.9 \pm 1.8$-fold increases, respectively, in basal channel activity as assessed by $Na^+$ currents at −50 mV (Fig. 1b, c). Substitution of neighboring residues (F556N and I559N) only showed about 2.5-fold increases of channel activity (Fig. 1c). Similar results were obtained for currents measured at other membrane potentials using a ramp protocol (Fig. 1d). Since leucine is similar in size to asparagine and alanine is smaller than asparagine, the L557N mutation would have negligible effect in the pore size and the A558N mutation would result in a smaller pore. Therefore, the increased channel activities observed in L557N and A558N mutants are presumably due to the removal of the hydrophobicity of L557 and A558, respectively. Biotinylation and immunofluorescence assays showed that the plasma membrane expression levels of these eight mutants are similar to that of the WT channel (Fig. 1e, f). We then performed noise analysis on whole-cell

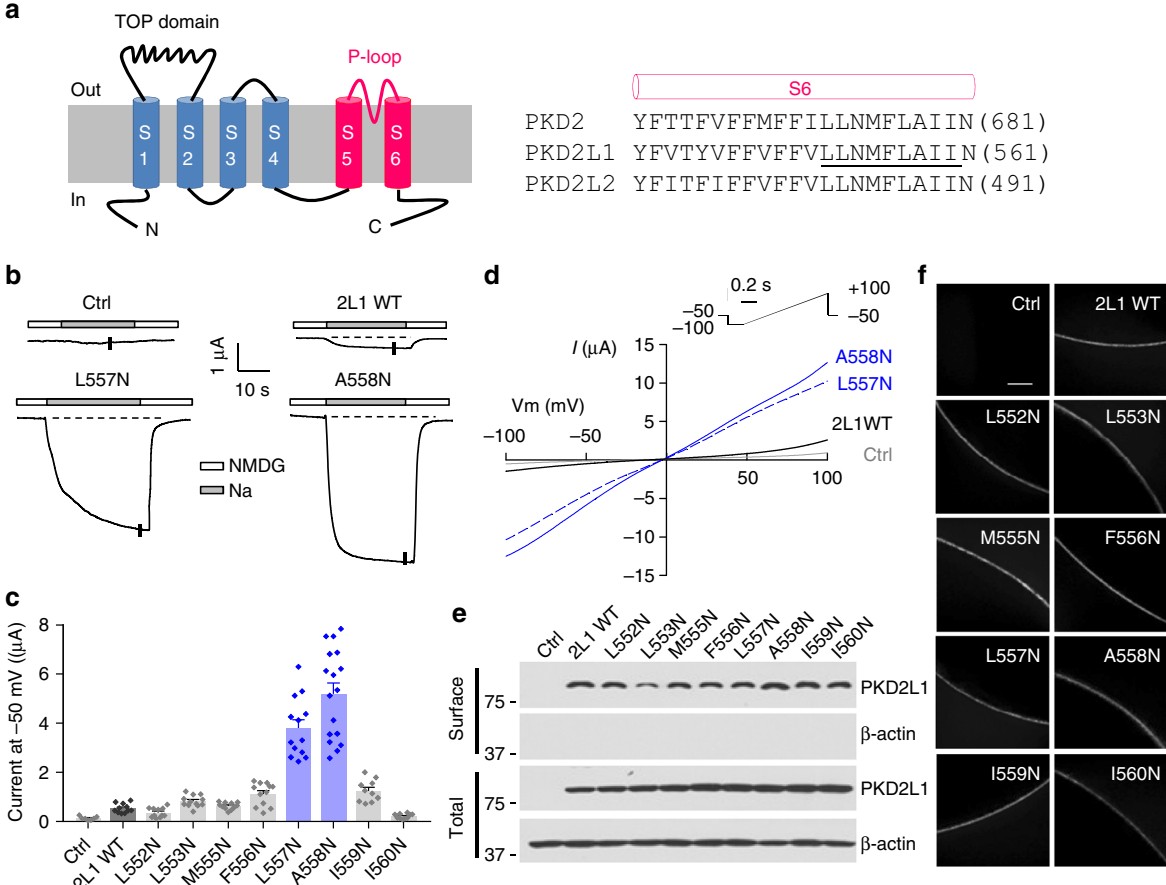

**Fig. 1** Identification of PKD2L1 hydrophobic gate in the S6 helix. **a** Left panel, membrane topology of polycystin channels. The TOP domain between S1 and S2 and the P-loop between S5 and S6 are indicated. Right panel, amino acid (aa) sequence alignment of S6 helix of human PKD2, PKD2L1 and PKD2L2. Each single underlined aa of PKD2L1 was mutated to asparagines (N). **b** Representative current traces obtained from oocytes injected with WT or a mutant (L557N or A558N) PKD2L1 cRNA, or water (Ctrl). Oocytes were voltage clamped at -50 mV in the presence of the extracellular solution containing (in mM) 100 $N$-methyl-D-glucamine (NMDG)-Cl, 2 KCl, 1 MgCl$_2$, 10 HEPES at pH 7.5 ("NMDG"), or Na-containing solution (equimolar Na$^+$ substituting NMDG) ("Na"). Dashed lines are the baselines from which plateau current values were determined. **c** Averaged currents as the difference between NMDG- and Na-containing solutions obtained at −50 mV, as in **b**, in oocytes expressing the single mutants, as indicated. For each mutant, currents were averaged from 10 to 17 oocytes from three batches. Data are presented as mean ± SEM. **d** Representative $I–V$ curves from Ctrl (water-injected oocytes), PKD2L1 WT, L557N or A558N expressing oocytes obtained at current points indicated by the vertical bars in the traces in **b**, using the indicated voltage ramp protocol. **e** Western blot of surface biotinylated and total PKD2L1 WT or indicated mutants. **f** Whole-mount immunofluorescence showing the oocyte surface expression of PKD2L1 WT or mutants from **e**. Scale bar, 50 µm

current traces obtained at −100 mV and found that both L557N and A558N mutants exhibited much higher open probability than WT PKD2L1 (0.65 ± 0.10 for L557N, 0.69 ± 0.12 for A558N, and 0.12 ± 0.03 for WT PKD2L1). Their single-channel currents were, however, comparable (13.5 ± 2.2 pA for L557N, 13.6 ± 2.4 pA for A558N, and 11.2 ± 1.6 pA for WT PKD2L1). The increased open probability without significant changes in single-channel current suggests that the helical-bundle-crossing (or physical constriction) mechanism plays a significant role in PKD2L1 gating in addition to the hydrophobic gating mechanism. Thus these data indicated that the consecutive L557 and A558 residues together form the hydrophobic gate in PKD2L1.

We then further replaced L557 with 12 additional amino acid residues of different hydrophobicity and/or sizes and found an interesting pattern of channel activity. Mutations to hydrophilic amino acids (E, D, N, Q, S, C, and T) all substantially increased channel activity (Fig. 2a), suggesting that the high hydrophobicity of L557 is an important factor to keep the PKD2L1 pore closed in the resting state. Mutations of L557 to hydrophobic amino acids (G, A, V, I, F, and W) together exhibited a reverse correlation

between the residue size and the channel activity (Fig. 2a), suggesting that PKD2L1 can also be partially gated by the physical constriction formed at site 557. Mutations at A558 in general also revealed a positive correlation between the hydrophilicity of site 558 and channel activity (Fig. 2b), with the exception that mutants A558T and A558G did not show any current increase at -50 mV as compared to the WT channel (Fig. 2b). Examination of the currents at other membrane potentials revealed that mutants A558T and A558G (as well as mutant A558S) in fact exhibit strong outward rectification and show around 6-fold increases in channel activity at +80 mV compared to the WT channel (Fig. 2c), although the mechanism underlying the rectification remains unknown. Removal of divalent cations from extracellular solutions had little effect on the rectification (Supplementary Figure 1), excluding the possibility of a blockade of inward currents by extracellular divalent cations as a cause of the rectification. Of note, the outward rectification was not observed in the corresponding L557 mutants (L557G, L557T and L557S) (Fig. 2d). The mutations at L557 and A558 did not affect PKD2L1 targeting to the surface membrane as shown by biotinylation and

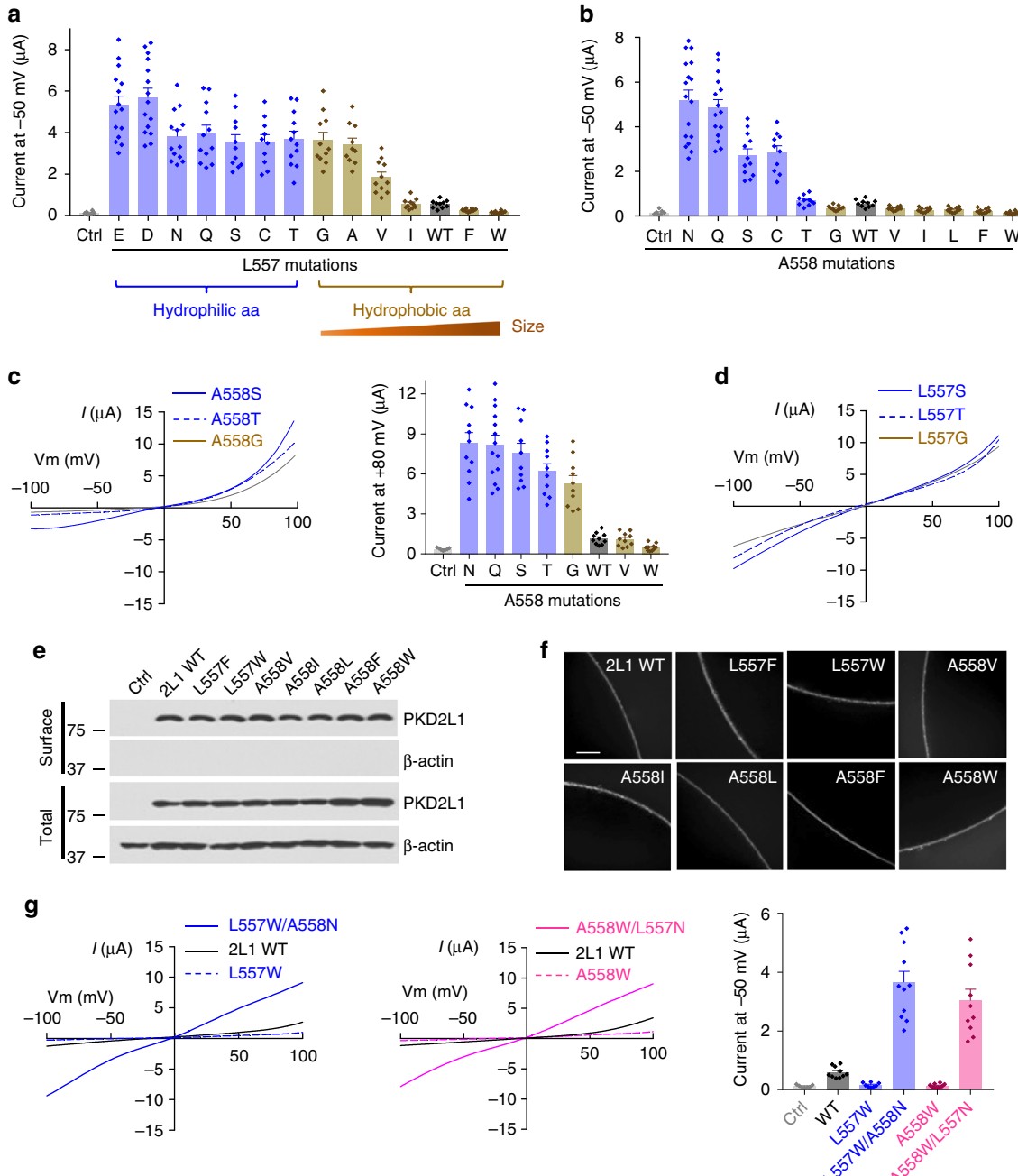

**Fig. 2** Characterization of mutants at PKD2L1 sites L557 and A558. **a** L557 was replaced with different aa, as indicated. Shown are averaged currents obtained under the same experimental conditions as in Fig. 1b ($n = 10$–15 oocytes from 3 batches). Data are presented as mean ± SEM. **b** A558 was replaced with different aa, as indicated, and averaged currents were obtained as in Fig. 1b ($n = 10$–17 oocytes from 3 batches). Data are presented as mean ± SEM. **c** Left panel, representative $I$–$V$ curves for three indicated A558 mutants generated under the same experimental condition as in Fig. 1d. Right panel, averaged outward currents at +80 mV for indicated A558 mutants ($n = 10$–14). Data are presented as mean ± SEM. **d** Representative $I$–$V$ curves for three indicated L557 mutants obtained as in Fig. 1d. **e** Western blot of surface biotinylated and total protein of PKD2L1 WT or indicated mutants. **f** Whole-mount immunofluorescence showing the oocyte surface expression of PKD2L1 WT or mutants from **e**. Scale bar, 50 μm. **g** L557 and A558 were replaced with W or N. Left panel, representative $I$–$V$ curves of PKD2L1 WT or indicated mutants obtained as in Fig. 1d. Right panel, averaged current at −50 mV for PKD2L1 WT or indicated mutants ($n = 8$–11). Data are presented as mean ± SEM

immunofluorescence data (Fig. 2e, f). These data indicated that the PKD2L1 pore is kept closed by two mechanisms: the hydrophobic barrier and physical constriction formed by the L557 and A558 gate residues. Disruption of either of them allows cation permeation through the pore.

A mutation might have multifaceted effects, e.g., altering inter- or intra-subunit interactions or affecting local or even global channel structure. In order to confirm that the observed effects of

mutations at L557 or A558 on channel activity are caused by changes in local hydrophobicity or pore size, we examined double mutants with each carrying two opposing mutations (one to hydrophilic N and the other to bulky hydrophobic W): L557N/ A558W and L557W/A558N. We found that both L557N/A558W and L557W/A558N mutants exhibit robust channel activity (Fig. 2g), indicating that the A558W or L557W LOF mutant (see Fig. 2a, b, g) is rescued by increasing hydrophilicity through the

L557N or A558N mutation. Taken together, these results strongly indicate that the adjacent L557 and A558 residues constitute the hydrophobic gate in PKD2L1 to prevent ion permeation in the closed state. Since only L677 was revealed to point to the central cavity and form the constriction in closed PKD2 structures (A678 points away from the cavity)[7–9], PKD2L1 might possess a slightly different ion-conducting pore from that of PKD2. We suspect that L557 and A558 in PKD2L1 together form the constriction in a way that both face the cavity but neither of them points directly to the cavity centre. Therefore, mutations at either L557 or A558

would alter the local pore hydrophobicity or size, thereby impacting the channel activity.

**Identification of hydrophobic gate residues in PKD2.** We wondered whether human PKD2 would possess the same gate residues as PKD2L1 since 10 amino acids around the PKD2L1 gate (LLNMF**LA**IIN) are identical in PKD2 (Fig. 1a). Overexpression of WT PKD2 in *Xenopus* oocytes produced no detectable current over background. We reasoned that a hydrophilic gate mutation would result in a constitutively open PKD2

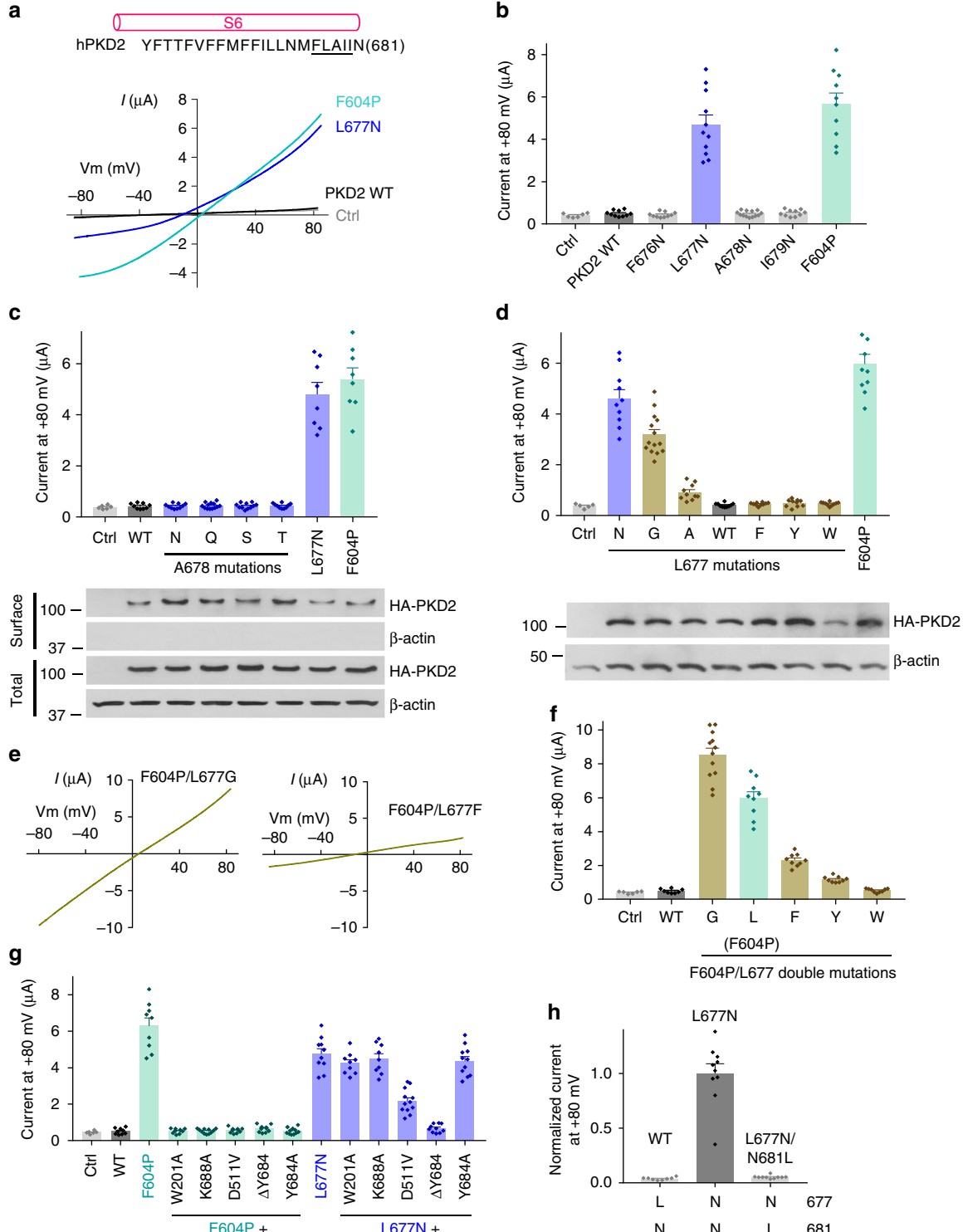

channel and produce detectable currents. We therefore replaced each of the hydrophobic residues in the F676-I679 fragment (corresponding to the gate-containing segment F556-I559 of PKD2L1) (Fig. 3a) with asparagine and found that only the L677N mutant (equivalent to the hydrophilic gate mutant L557N in PKD2L1) exhibits substantially increased channel activity, comparable to that of the GOF mutant F604P that we reported recently[5] (Fig. 3a, b). Residue A678, which corresponds to PKD2L1 gate residue A558, did not conduct appreciable current when replaced with N (Fig. 3b). Replacement of A678 with other hydrophilic residues (Q, S, and T) did not produce any detectable current either (Fig. 3c), suggesting that A678 is not part of the PKD2 gate. Surface biotinylation and immunofluorescence showed that the plasma membrane targeting of mutants A678N, -Q, -S and -T, L677N, and F604P is similar to that of the WT protein (Fig. 3c and Supplementary Figure 2). We next focused on L677 by substituting it with different amino acids and found that replacing L677 with N or G, but not with bulky hydrophobic residues (F, Y, and W), indeed produces robust currents (Fig. 3d). Replacing it with the smaller hydrophobic residue alanine partially opened the pore (Fig. 3d). We also found that in the F604P mutant, replacing L677 with the small residue G further increases the activity while replacing it with large hydrophobic residues (F, Y, and W) substantially attenuates activity (Fig. 3e, f). These functional data support the conclusion that L677 but not A678 serves as a single-residue gate of PKD2. The functionally identified gate residue (L677) is in agreement with the known structures of PKD2[7–9], where L677 forms a physical constriction within the pore, and thus validates our methodology.

Although both F604P and L677N mutations resulted in constitutively active PKD2 channels, we assumed that they open the pore in different ways, i.e., the F604P mutation induces a series of conformational changes in the PKD2 channel that ultimately leads to gate movements and pore opening (see below), whereas isosteric polar substitution L677N opens the pore by directly increasing the gate's hydrophilicity. In other words, the F604P mutant would be in an activated conformation while L677N would still be in a resting conformation. Interestingly, we found that the channel activity of mutant F604P is abolished by each of the following mutations: W201A, K688A, D511V, Y684A and ΔY684 (Y684 residue deletion) (Fig. 3g). These mutations may have prevented conformational changes required for channel activation[30]. Of note, D511V and ΔY684 are two naturally occurred mutations found in human ADPKD patients[31,32]. In contrast, the channel activity of mutant L677N was not very affected by these mutations, except for ΔY684 (Fig. 3g), which supports our assumption that mutant L677N is in a different state from that of mutant F604P and gains channel function due to removal of hydrophobicity at L677.

In the PKD2 structures[7–9], the four N681 residues establish another narrow pore constriction (with a radius of 3.1 Å) about one helical turn below the L677 residues that form a constriction with a radius of 2.5 Å[9]. However, because of its hydrophilicity, we think that N681 is not as effective as L677 to block ion permeation. This is supported by the fact that the isosteric mutation L677N alone was sufficient to open the pore and resulted in a GOF PKD2 channel. In addition, replacing N681 with isosteric hydrophobic L in the open-gate channel, mutant L677N, closed the channel (Fig. 3h), presumably because a new hydrophobic gate is formed by L681 in L677N/N681L. In other words, the PKD2 pore is conductive if both constrictions are formed by hydrophilic N (i.e., N677 and N681 in L677N), whereas it is closed if either constriction is formed by hydrophobic L (i.e., L677 in WT or L681 in L677N/N681L) (Fig. 3h). Taken together, our data indicates that in the PKD2 WT channel, the pore is kept closed in the resting state by the hydrophobic residue L677.

**PKD2 GOF gate mutants in vivo**. We next examined the function of PKD2 GOF gate mutants in zebrafish embryos in which knockdown of the PKD2 through injection of morpholino oligonucleotide (MO) resulted in tail curvature and pronephric cysts[33]. Although it remains unclear as to how tail curvature is induced by PKD2 knockdown, the presence of pronephric cysts is reminiscent of the renal cysts in human ADPKD patients and can be observed under regular light microscopy from 3 days post-fertilization (dpf) onward[34]. Injection of 1 ng PKD2 MO into one-cell zebrafish embryos within 1 h post-fertilization resulted in tail curling to various degrees at 3 dpf, which was divided into four groups: normal (no tail curling), moderate (significant curling, < 90°), substantial (90°–180°) and severe (>180°) (Fig. 4a). Co-injection with mRNAs encoding human PKD2 WT or mutant L677G but not L677W significantly rescued the tail curling phenotype and showed similar rescuing effects as GOF mutant F604P[5] (Fig. 4b). Furthermore, double mutant F604P/L677G gave rise to a more pronounced rescuing effect than either the F604P or L677G single mutant (Fig. 4b), which is in agreement with the larger currents observed in F604P/L677G as compared to that of F604P or L677G (Fig. 3d, f). When pronephric cysts were counted by microscopy (Fig. 4c), about 20% of larval fish with PKD2 MO injection at 3 dpf had cystic pronephros (Fig. 4d). Co-injection of PKD2 L677G or F604P mRNA, but not L677W or WT PKD2 mRNA, significantly rescued the cystic phenotype (Fig. 4d). Similar to the tail curling assay, the F604P/L677G double mutant showed a more pronounced rescuing effect than the corresponding single mutants. The reason why human PKD2 GOF mutants did not completely rescue pronephric cysts (Fig. 4d) remained unclear. It may be due to either functional differences between human and zebrafish PKD2, which only share 63%

**Fig. 3** Identification and characterization of PKD2 hydrophobic gate. **a** Each single underlined aa of human PKD2 was mutated to N. Representative *I–V* curves were obtained in oocytes expressing WT or a mutant PKD2, as indicated, in the presence of the divalent free Na-containing solution (in mM): 100 NaCl, 2 KCl, 10 HEPES at pH 7.5. The gain-of-function mutant F604P (in the S5 helix) serves as a positive control and water-injected oocytes as a negative control (Ctrl). **b** Averaged currents at +80 mV obtained under the same experimental conditions as in **a** in expressing or control oocytes, as indicated. Currents were averaged from 10–13 oocytes from at least three batches. Data are presented as mean ± SEM. **c** Upper panel, averaged currents obtained as in panel a for PKD2 WT, A678 mutants, L677N or F604P ($n = 8$–13). Data are presented as mean ± SEM. Lower panel, western blot of surface biotinylated and total protein of PKD2 WT or indicated mutants. **d** Upper panel, the PKD2 L677 was replaced with different aa as indicated. Shown are averaged currents obtained as in **a**, with F604P as a positive control ($n = 10$–14). Data are presented as mean ± SEM. Lower panel, western blot showing total protein of PKD2 WT or indicated mutants present in the injected oocytes. **e** Representative *I–V* curves for F604P/L677 double mutants, as indicated, under the same condition as in **a**. **f** Averaged currents at +80 mV recorded from oocytes injected with cRNA of the indicated PKD2 WT or F604P/L677 double mutants ($n = 8$–12). Data are presented as mean ± SEM. **g** Averaged currents at +80 mV obtained from oocytes expressing PKD2 WT, or indicated single or double mutants ($n = 9$–12). Data are presented as mean ± SEM. **h** Averaged currents at +80 mV obtained from oocytes expressing PKD2 WT, L677N or L677N/N681L mutant. Data are presented as mean ± SEM

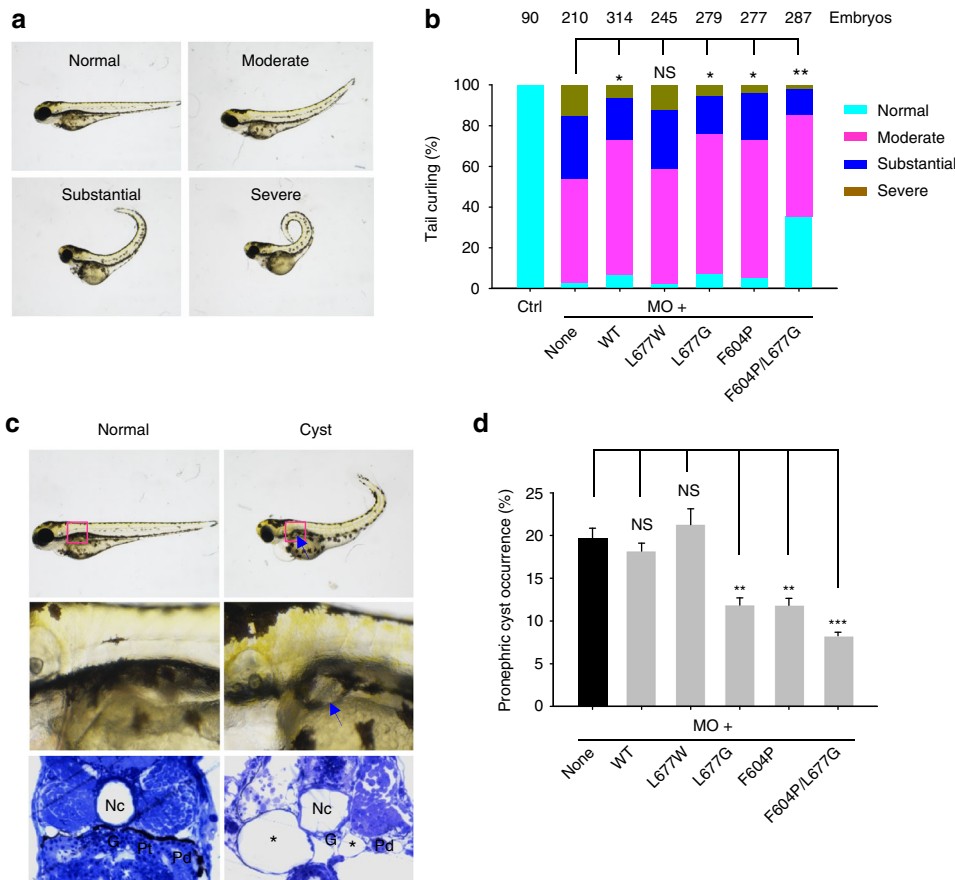

**Fig. 4** Rescue of PKD2-associated zebrafish phenotypes by PKD2 gate mutants. **a** Representative zebrafish embryos showing different severities in tail curling induced by PKD2 MO knockdown. Normal (no curvature), Moderate (significant curvature, <90°), Substantial (90°–180°), and Severe (>180°). **b** Averaged percentages of embryos with Normal, Moderate, Substantial and Severe tail curling at 3dpf. Embryos were co-injected with 2.5 ng PKD2 MO together with none or 100 pg mRNAs of WT or mutant PKD2, as indicated. Ctrl, uninjected embryos. Data were from three independent experiments with the indicated total numbers of embryos. Statistic significance was determined by the $\chi^2$ test. NS, no significance; *$P < 0.05$; **$P < 0.01$. PKD2 F604P mutant serves as a positive control. **c** Water-injected (Normal) and PKD2 MO-injected (Cyst) zebrafish embryos at 3 dpf showing curly tail and pronephric cyst formation (arrows) which is confirmed by a histologic section that also displayed dilated pronephric tubules (asterisks). G, glomerulus; Pt/Pd, pronephric tubule/duct; Nc, notochord. **d** Averaged percentages of embryos exhibiting pronephric cysts under the same conditions as those in **b**. Data are presented as mean ± SEM. Statistic significance was determined by Student's $t$-test. NS, no significance; **$P < 0.01$; ***$P < 0.001$

amino acid identity, or lower channel activities of GOF PKD2 mutants in zebrafish pronephric cells compared to those in *Xenopus* oocytes. Taken together, these data indicate that L677G acts as a GOF mutation in vivo in zebrafish.

**Allosteric gating mechanisms in PKD2.** To elucidate the activation mechanism of PKD2, we resolved a structure of the PKD2 F604P GOF mutant[5], which our data suggest to be trapped in an activated state. We employed a truncated human PKD2 channel, E53-D792, in our structural study since it was shown to exhibit enhanced biochemical stability and structural homogeneity over the full-length protein[9]. When expressed in *Xenopus* oocytes, both the full-length and truncated PKD2 showed constitutive channel activities in the presence of the F604P mutation (Supplementary Figure 3). We determined a cryo-EM structure of the PKD2 F604P mutant to an overall resolution of 3.5 Å (Supplementary Figures 4-7, Supplementary Table 1). Similar to the WT channel, the F604P mutant map showed unambiguous density for the channel core, but no extra densities to account for the cytoplasmic N- and C-termini. Overall, the F604P mutant shared a similar tetrameric architecture and domain arrangements with WT PKD2 (Fig. 5a, b). No significant structural change was found in the TOP domain, ion selectivity filter or S1-S4 domain

(Supplementary Figure 8). However, the F604P mutation resulted in a kink in S5 and a displacement of the bottom half of S5 below 604 F/P (Fig. 5c), which leads to expansion of the activation gate that likely underlies the GOF phenotype (Fig. 5d, e).

The S4–S5 linker was previously proposed to participate in allosteric gating of TRP channels and contains numerous pathogenic GOF or LOF mutations[35]. Interestingly, when comparing the PKD2 WT and F604P structures, the S4-S5 linker was shown to exhibit a notable conformational change: the N-terminal end of the linker in F604P moved outward, which led to a dramatic displacement of the cationic residue R581 that is normally conserved in the TRPP family (Fig. 5f). Indeed, the R581A substitution in PKD2 substantially reduced F604P channel activity (Fig. 5g) and the corresponding K461A substitution in PKD2L1 significantly reduced its $Ca^{2+}$-induced currents in *Xenopus* oocytes[21] (Fig. 5h). These results further support the involvement of this region in PKD2 and PKD2L1 allosteric gating. Interestingly, the outward movement of the N-terminal end of the S4-S5 linker was also observed in TRPML1 in a proposed pre-open state, but not in a closed state (Supplementary Figure 9)[36], indicating that the S4-S5 linker may play a similar role in the gating of PKD2 and TRPML1.

We previously observed a π-helix, formed by ${}^{668}$MFFIL${}^{672}$, in the middle of S6 in PKD2[9] and such a π-helix was also observed

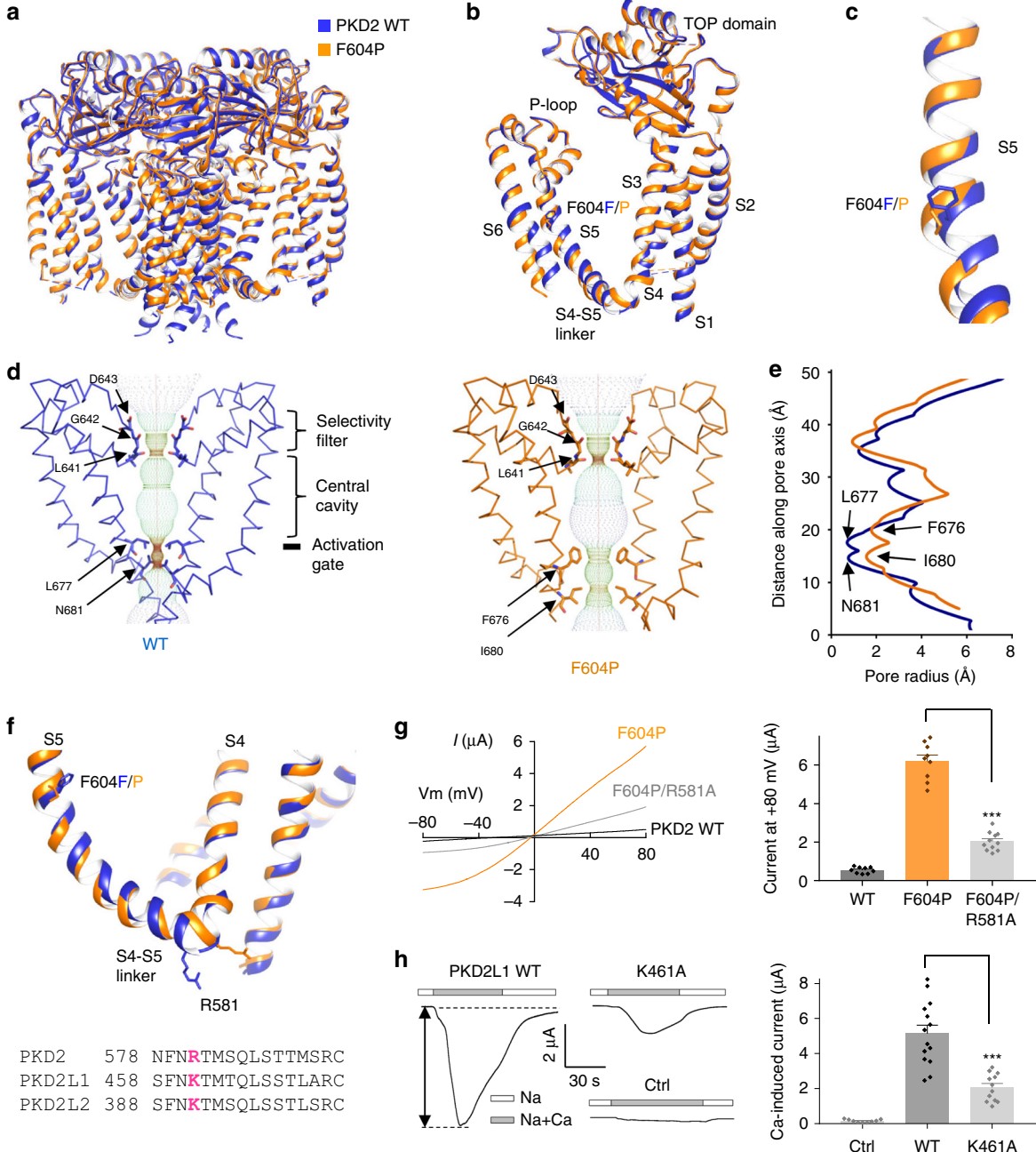

**Fig. 5** Comparison of structure of PKD2 F604P with that of WT channel. **a**, **b** Side view of superposition of tetrameric (**a**) and subunit (**b**) structures of PKD2 F604P (orange) and WT (blue). **c** Comparison between S5 helix in PKD2 F604P and WT channels. The F604 and P604 residues are shown. **d** Solvent-accessible pathway along the pore mapped using the HOLE program for PKD2 WT and F604P structures. Residues forming the selectivity filter and lower constriction points are indicated. **e** Comparison between the pore radii, calculated with the program HOLE, for the PKD2 WT and F604P structures. **f** Upper panel, comparison between the S4-S5 linker in the PKD2 WT and F604P structures. The R581 at the beginning of the linker is shown. Lower panel, sequence alignment of S4-S5 linker of human PKD2, PKD2L1 and PKD2L2. The conserved cationic residues were highlighted in bold and magenta. **g** Left panel, Representative *I-V* curves for PKD2 WT, single mutant F604P and double mutant F604P/R581A obtained under the same condition as in Fig. 3a. Right panel, averaged currents at +80 mV recorded from oocytes injected with cRNA of PKD2 WT or mutant F604P or F604P/R581A. Data are presented as mean ± SEM. ***P < 0.001 by Student's *t*-test. **h** Left panel, representative whole-cell current traces obtained from control oocytes (Ctrl, water-injected) or oocytes injected with cRNA of human PKD2L1 WT or mutant K568A. Oocytes were voltage clamped at −50 mV and currents were recorded using Na-containing solution (see Fig. 1b) without (Na) or with addition of 5 mM CaCl₂ (Na + Ca). The Ca-activated peak current, indicated by the double arrowed line, was measured to assess PKD2L1 channel activation. Right panel, averaged Ca-activated peak currents from oocytes expressing PKD2L1 WT or K461A mutant or water-injected oocytes (Ctrl). Data are presented as mean ± SEM. ***P < 0.001 by Student's *t*-test

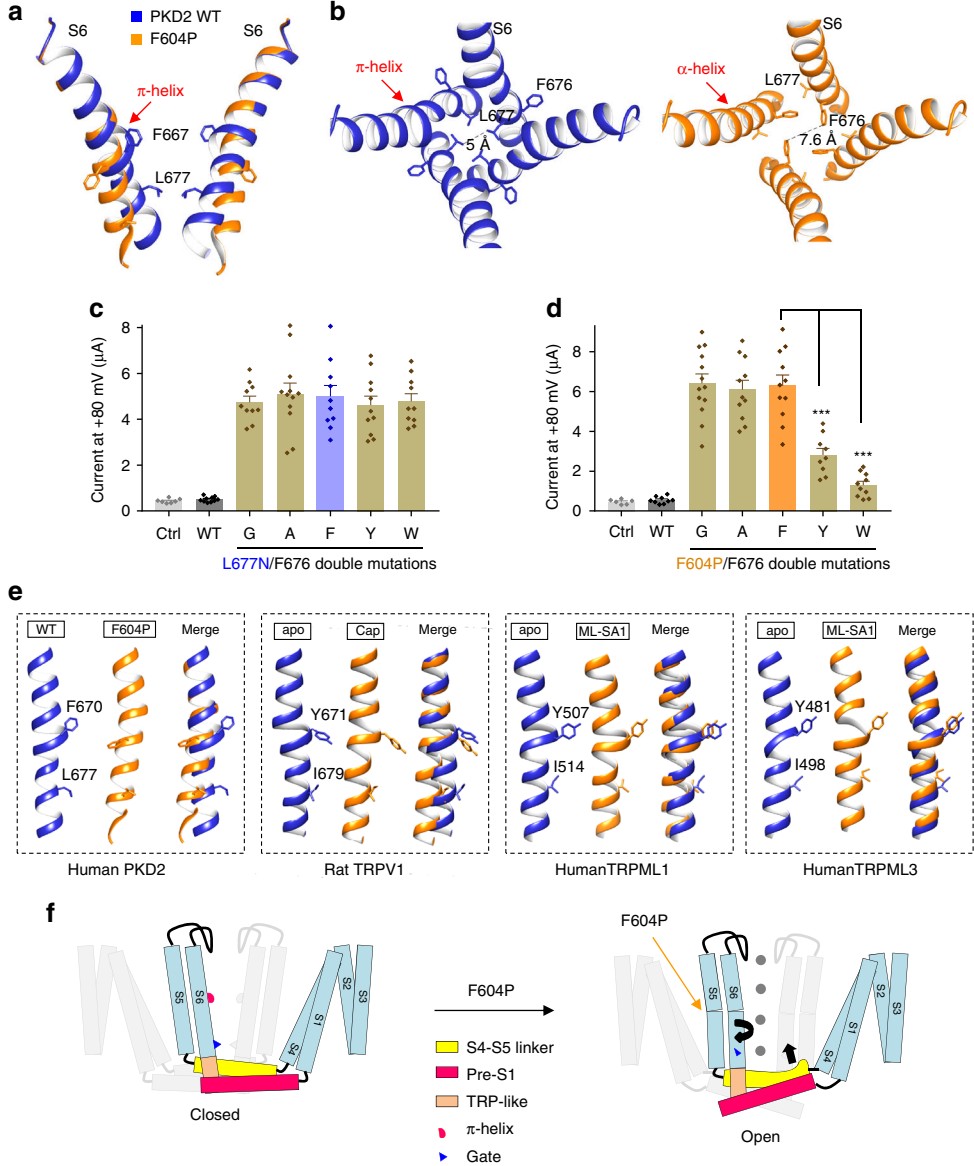

**Fig. 6** PKD2 S6 conformational changes induced by F604P mutation. **a** Side view of superposition of S6 in the PKD2 WT (blue) and F604P (orange). The π-helix in the WT structure is indicated by an arrow. The F667 residue, located in the π-helix, and gate residue L677 are shown. **b** Top view of the pore lined by S6 in the PKD2 WT and F604P structures showing twisting and bending movements of distal S6 induced by the transition of π-helix to α-helix. **c**, **d** Averaged currents at +80 mV obtained under the same experimental conditions as in Fig. 3a in oocytes injected with cRNA of PKD2 WT, L677N/F676 (**c**) or F604P/F676 (**d**) double mutant. Currents were averaged from 10–13 oocytes. Ctrl, water-injected oocytes. Data are presented as mean ± SEM. Statistic significance was determined by Student's t-test. ***P < 0.001. **e** Comparison between S6 in the closed (blue) and open (orange) states of PKD2, TRPV1, TRPML1 or TRPML3. The aromatic residue in the π-helix and hydrophobic gate residue were shown for each channel. PKD2 WT (PDB: 5T4D); TRPV1 apo (unliganded and closed state, PDB: 3J5P), Cap (capsaicin-bound state, PDB: 3J5R); TRPML1 apo (PDB: 5WJ5), ML-SA1 (agonist ML-SA1-bound state, PDB: 5WJ9); TRPML3 apo (PDB: 6AYE), ML-SA1 (6AYF). **f** Proposed mechanistic model for PKD2 activation induced by F604P mutation. The black arrows indicate proposed movements during the activation

in TRPV1, -A1, -ML1, -ML3, -M4 and NOMPC[36–40] (Supplementary Figure 10). By comparing the TRPV2 and TRPV1 structures, transition of this π-helix to a canonical α-helix was previously hypothesized to mediate TRPV2 gating[41]. Here, by directly comparing S6 of PKD2 WT and F604P mutant, we found a transition of the π-helix in WT PKD2 to the canonical α-helix in F604P mutant. This transition in PKD2 induced a combined twisting and bending motion of the distal S6 below the π-helix, resulting in twisting and splaying of gate residue L677 (Fig. 6a) that expands the pore from 5 to 7.6 Å (Fig. 6b). The pore dilation observed in the F604P mutant is similar to that of TRPV1 (from

5.3 to 7.6 Å by capsaicin)[42], TRPML1 (from 5.6 to 8.4 Å by ML-SA1)[43] and TRPML3 (from 5.3 to 10.4 Å by ML-SA1)[44], despite the fact that they result from different gating motions (see Discussion and Fig. 6e). In contrast to the dramatic changes of distal S6, the proximal S6 above the π-helix remained rather stationary in PKD2 (Fig. 6a), suggesting that the transition from π- to α-helix alone is sufficient to induce pore opening and channel activation. The twisting motion in PKD2 not only induced the gate residue L677 to rotate and point away from the pore center, but also resulted in turning of the precedent residue F676 towards the pore (Fig. 6b). In other words, the F676 points

away from the pore in the PKD2 WT and mutant L677N (resting state), but faces the pore in the F604P mutant (activated state) as a result of the twisting. We next examined the functional consequences of the rotation of F676. In the presence of the L677N mutation, replacing F676 with either a large hydrophobic residue (Y or W) or with a small residue (G or A) had little effect on PKD2 channel activity (Fig. 6c), consistent with the WT PKD2 structure where F676 points away from the pore. In mutant F604P, however, the channel activity was significantly blocked by substitution of F676 with Y or W, but not with G or A (Fig. 6d), consistent with the observation that residue F676 now points towards the pore. Because large hydrophobic residue substitutions at L677 also significantly blocked F604P channel activity (Fig. 3f), these data suggest the presence of a double-residue gate F676-L677 when PKD2 is in an activated state.

Taken together, our structural and functional data suggested a gating mechanism of PKD2 whereby a transition of π-helix to α-helix in the middle of S6 induces twisting and splaying of the hydrophobic gate residue that leads to expansion of the pore and channel activation. Of note, PKD2 may be activated by as-yet unidentified endogenous ligand(s), which could involve gating mechanisms that are distinct from the one associated with the F604P mutation.

## Discussion

The gate of an ion channel is defined as the part of the ion conducting pore that effectively prevents ion passage when the channel is in a closed state and undergoes conformational changes to allow ion permeation upon channel activation. TRPV1 structures revealed two physical constrictions along the ion conducting pore, one formed by the hydrophilic G643 residue in the selectivity filter (also called upper gate) and the other by hydrophobic I679 residue at the lower gate (also called activation gate)[37,42]. Interestingly, agonist capsaicin only induced widening of the activation gate, while resiniferatoxin (RTX) and spider double-knot toxin (DkTx) together induced widening of both the activation gate and selectivity filter[37], suggesting that the hydrophobic I679 may serve as a primary gate and the hydrophilic G463 as a secondary gate involved in activation by only a subset of TRPV1 agonists, e.g., RTX/DkTx. All reported PKD2 structures revealed two similar constrictions along the pore[7–9]. Compared to the PKD2 WT structure captured in a closed state, the constitutively active F604P mutant showed no significant change in the selectivity filter (Supplementary Figure 8b), suggesting that the selectivity filter does not act as a gate in F604P mutation-induced activation. It remains to be determined whether it could serve as a secondary gate in the presence of some unidentified agonists. The activation gate in PKD2 was previously proposed to be constituted by L677 and N681, based on the observation that each of them forms similar physical constriction points[7–9]. However, the activation gate of PKD2 has not yet been functionally characterized. Our data now indicated that the PKD2 activation gate is primarily defined by the hydrophobic L677 and that the hydrophilic N681 is less effective than L677 in preventing ion permeation. Interestingly, all resolved structures of TRP channels in closed states to date revealed that pore diameters at gates are all within 5–6 Å (Supplementary Figure 10), which is comparable to the sizes of hydrated $Na^+$ (5.5 Å) and $K^+$ (4.6 Å) ions. Therefore, the effect of physical constriction alone would be insufficient to completely prevent ion permeation in the closed state, which explains why gate residues in TRP channels have all converged to be hydrophobic during evolution. Taken together, activation gates in TRP channels block ion permeation in closed states via two synergistic mechanisms: hydrophobic barrier and physical constriction.

Although PKD2L1 and PKD2 are close homologues (54% overall amino acid identity) and share 10 identical amino acids around the lower gate, we found that they adopt distinct gate apparatuses, i.e., the single-residue gate (L677) in PKD2 vs. the double-residue gate (L557-A558) in PKD2L1. Interestingly, PKD2 is highly conserved during evolution and its orthologs have been identified in lower organisms such as yeast, *Drosophila* and *C. elegans*[45,46]. PKD2L1, meanwhile, is currently only found in higher organisms. We speculate that PKD2L1 may first arise via duplication of the PKD2 gene, and during revolution residues outside the pore diverged from PKD2, which necessitates a concerted accommodating structural changes at the pore (e.g., gate residue twisting), resulting in a double-residue gate. We previously also identified double-residue gates in TRPV5 and TRPV6[28]. It would be interesting to examine whether and how these double-residue gates are related to biophysical properties and physiological function of PKD2L1, TRPV5, or TRPV6.

The gating of an ion channel may involve complex conformational changes that ultimately lead to movements of the gate and pore opening. To date, at least three generic modes of gating motion have been proposed[47]. The first one consists of a rigid-body iris-like rotation of the pore-lining helices around the pore axis (e.g., in MscS channel[48]). In the second mode, a hinge in the middle of pore-lining helices allows the distal helices to kink at the hinge to open the gate upon activation. A conserved glycine residue in many $K^+$ channels has been identified to act as such a hinge[49]. The third mode involves a hinge together with additional retraction of an aromatic gate residue which is found in several $K^+$ channels, including KirBac1.1 and KirBac3.1[50,51]. In this study, we revealed a gating motion in PKD2 that involves the transition of a π-helix to α-helix in the middle of the pore-lining S6, resulting in a combined twisting and bending motion of the distal S6, which led to twisting and splaying of gate residue L677 and pore opening (Fig. 6e). Such a transition was initially proposed as a gating mechanism in TRPV2[41]. It should be kept in mind that a truncated PKD2 mutant E53-D792 was used in our structural studies while the full-length PKD2 was used in the functional assays. The structure of this PKD2 truncated mutant might be different from that of full-length PKD2, which may lead to misinterpretation of our functional data in the structural context. This π-helix structure is present in several other TRP channels, including TRPV1, TRPA1, TRPML1, TRPML3, NOMPC, TRPM4, and TRPV6, but not in TRPV2 or TRPM8 (Supplementary Figure 10), suggesting that the π-helix may represent a shared gating element for many TRP channels, similar to the gating hinge (conserved glycine residue) in $K^+$ channels[49]. Interestingly, the most recently published TRPV6 structures in closed and open states revealed an α-helix to π-helix transition during TRPV6 activation[52], which is opposite to the π-helix to α-helix transition associated with PKD2 activation. Of note, the π-helix could function differently in the gating of individual TRP members. In TRPV1, the π-helix more likely acts as a hinge to allow bending of the distal S6 upon activation induced by agonist capsaicin (Fig. 6e), analogous to the glycine hinge in $K^+$ channels mentioned above. In TRPML1 and TRPML3, the π-helix does not seem to play a critical role in agonist ML-SA1-induced channel activation, which is accomplished by displacement of the entire S6 helix (Fig. 6e). The diverse roles of π-helices possibly reflect the fact that TRP channels are polymodal receptors responding to multiple stimuli. Future efforts are needed to resolve more open-state structures in the presence of different stimuli to elucidate the dynamic functions of the π-helix in the gating of TRP channels.

To date, several studies have reported that cytoplasmic $Ca^{2+}$ can act as an agonist of PKD2 at low concentrations[6,20,53–55]. It is highly possible that PKD2 also has other unidentified agonists. These agonists, including $Ca^{2+}$, may utilize gating mechanisms

distinct from that associated with the F604P mutation. Cytoplasmic $Ca^{2+}$ was reported to increase the probability of PKD2 opening at low concentrations while inhibiting PKD2 opening at high concentrations[20,53,54]. Consistent with the two modes of $Ca^{2+}$ regulation, two distinct PKD2 structures in complex with $Ca^{2+}$ were determined. The more open structure has one $Ca^{2+}$ ion bound below the selectivity filter while the other structure, hypothesized to be in an inactivated state, is bound with multiple $Ca^{2+}$ ions[7]. Interestingly, comparison of the two structures revealed a local conformational change in the middle of S6, resulting in a prominent movement of the L677 and pore opening (Supplementary Figure 11), which is distinct from the movement induced by the F604P mutation (Fig. 6a, b). Both the PKD2 F604P and WT structures with one $Ca^{2+}$ ion bound showed a pore diameter of 7.6 Å at the lower gate (Fig. 6b, Supplementary Figure 6b), whereas a functional study using organic monovalent cations of various sizes predicted the PKD2 pore diameter to be around 11 Å[56]. It could be that the PKD2 channel in the functional study resides in a state that is different from those revealed by the structural studies. Although it remains unclear how PKD2 is regulated by $Ca^{2+}$, an EF-hand domain containing two helix-loop-helix (or EF-hand) motifs was identified in the PKD2 C-terminus for $Ca^{2+}$ sensing[57]. Interestingly, the number and location of EF-hand motifs were shown to be important for the regulation of PKD2 function by $Ca^{2+}$[53]. In human PKD2, the first EF-hand motif lacks $Ca^{2+}$ binding capability due to loss of four evolutionarily conserved $Ca^{2+}$-coordinating residues in the loop region. In lower species such as sea urchin, however, both EF-motifs are functional and act cooperatively, resulting in a much higher affinity for $Ca^{2+}$[53,58]. The addition of the four missing $Ca^{2+}$-binding residues to the first EF-hand motif in human PKD2 restored a functional $Ca^{2+}$ binding site and enhanced the regulation of PKD2 channel function by $Ca^{2+}$[53]. At low $Ca^{2+}$ levels, the binding of $Ca^{2+}$ to an EF-hand may lead to conformational changes in the PKD2 C-terminus, which may then propagate to the lower gate and result in pore opening. Meanwhile, at high $Ca^{2+}$ levels, a $Ca^{2+}$ binding site(s) outside the PKD2 C-terminus with low $Ca^{2+}$ affinity, such as the selectivity filter and outer pore regions, may start to bind with $Ca^{2+}$ and inactivate the PKD2 channel.

Our current functional and structural studies demonstrated that PKD2 can form a homotetrameric channel in heterologous expression systems. In physiological conditions, PKD2 is generally hypothesized to function in a complex with PKD1 in primary cilia of renal epithelial cells. This concept was challenged by a recent report showing that PKD2 forms an essential ion channel independent of PKD1 in the primary cilium of the renal collecting duct epithelium[6]. This ciliary PKD2 channel was shown to preferentially conduct $Na^+$ and $K^+$ over $Ca^{2+}$. Our structures of PKD2 WT and F604P mutant provided guidance for mutagenesis studies to explore the structural basis underlying PKD2 preference for monovalent cations over $Ca^{2+}$. Although exhibiting low $Ca^{2+}$ conductance, PKD2 was found to be inhibited by extracellular $Ca^{2+}$ and potentiated by intraciliary $Ca^{2+}$[6]. The positive regulation by intraciliary $Ca^{2+}$ could be dependent on the C-terminal EF-hand domains, which are not resolved in our PKD2 structures. The structure of full-length PKD2 would be valuable to provide insights into the $Ca^{2+}$-dependent regulation of PKD2 activity.

It is currently unclear how exactly the F604P mutation induces the transition of π-helix to α-helix and consequently pore opening. Superimposition of WT and F604P structures revealed no notable changes in the TOP domain, S1–S4 helices and selectivity filter (Supplementary Figure 8), suggesting that these parts are not involved in F604P-induced channel activation. Interestingly, we observed dramatic changes in the N-terminus of the S4–S5 linker induced by the PKD2 F604P mutation (Fig. 5f), and

intramolecular proximity among the S4–S5 linker, pre-S1 and TRP/TRP-like domain in structures of TRPV1, TRPV2, TRPV6, TRPA1, and NOMPC[37–39,41,59]. We thus proposed a gating mechanism of PKD2 that depends on interactions/coupling among S4–S5 linker, pre-S1 and TRP-like domain (see Fig. 6f). In this gating scenario, the mutation F604P creates a kink in the middle of S5 and results in a bending of the bottom half of S5, which induces an outward movement of N-terminus of the S4–S5 linker. This displacement in the S4–S5 linker is then coupled by synchronized movements of the pre-S1 and TRP-like domain, which together induce energy release in the π-helix in S6, triggering a transition to an α-helix. Of note, the TRP-like domain in PKD2 is a continuation of the S6 helix and is almost perpendicular to the membrane whereas the TRP/TRP-like domain in TRPV1, TRPV2, TRPA1, or TRPV6 is largely parallel to the membrane and perpendicular to S6[9,37,38,41,59].

Since PKD2 and PKD2L1 share a conserved S6 sequence and overall tetrameric architecture with other TRP channels, our strategy of generating GOF mutants via gate residue substitutions is likely generally applicable to other TRPs without known agonists, e.g., TRPC1. Gene knockout animals were reported to show no expected phenotypes via activation of compensatory networks[60]. Here, we identified a series of GOF gate mutants with channel activity increased to various extents, which would allow generating alternative GOF animal models through knock-in technology to study the in vivo function of TRP members, including PKD2L1 and PKD2. This approach is encouraged by our zebrafish experiments, which showed that the PKD2 GOF gate mutants are functional in vivo. However, knock-in of a strong GOF mutant may also be deleterious, e.g., due to cytotoxic cation overload.

In summary, we have identified hydrophobic gates in PKD2L1 (L557-A558) and PKD2 (L677) in the distal part of the S6 helix and obtained a series of GOF and LOF gate mutants. PKD2 GOF gate mutants showed rescuing effect on PKD2 knockdown-induced tail curling and pronephric cysts in embryonic zebrafish. We also resolved a structure of the PKD2 F604P mutant by cryo-EM and identified a PKD2 gating mechanism involving a π- to α-helix transition within S6, which induces twisting and splaying of the gate residue and consequently channel activation. Our findings provided insights into the fundamental channel function of PKD2, which is essential for developing therapeutic interventions for ADPKD.

## Methods

**Plasmids, mutants and antibodies**. Human PKD2L1 cDNA (accession #: NM_016112) was inserted into vector pCHGF[61] for efficient expression in *Xenopus laevis* oocyte. Flag tag was then added before the N-terminus of the PKD2L1 for immunodetection. HA-tagged human PKD2 (NM_000297) plasmid was constructed as we previously reported[5]. Mutagenesis was carried out using QuikChange Lightning Site-Directed Mutagenesis kit (Agilent Technologies, La Jolla, CA) and confirmed by sequencing. Sequences of primers used in this study are included in Supplementary Table 2. Rabbit FLAG (D-8, 1:2000), HA (Y-11, 1:2000) and mouse β-actin (C-4, 1:4000) antibodies were from Santa Cruz Biotechnology (Santa Cruz, CA) for immunoblotting. Secondary antibodies against rabbit (NA934-1ML, 1:2000) or mouse (NA931-1ML, 1:2000) IgG were from GE Healthcare (Waukesha, WI).

**Xenopus oocyte expression**. Capped synthetic RNA of human PKD2L1 and PKD2 were synthesized by in vitro transcription using mMESSAGE mMACHINE kit (Ambion, Austin, TX) and injected at 25 ng/oocyte into *Xenopus* oocytes prepared as described[62]. Briefly, isolated oocytes from *Xenopus laevis* were defolliculated with 2 mg/ml collagenase at room temperature (RT) for 2 hours (h). Dissociated oocytes at stage V–VI were manually sorted out and maintained at 18 °C for at least 3 h before mRNA injection. Equal volumes of water were injected into oocytes as negative controls. Experiments were performed 1–3 days following RNA injection. This study has been approved by the Ethical Committee for Animal Experiments, University of Alberta, and was conducted according to the Guidelines for Research with Experimental Animals (University of Alberta) and for the Care and Use of Laboratory Animals (NIH Guide, revised in 1996).

**Two-electrode voltage clamp**. Two-electrode voltage clamp experiments using oocytes were carried out as previous[61]. Briefly, two electrodes made of capillary pipettes (Warner Instruments, Hamden, CT) penetrating an oocyte were filled with 3 M KCl to obtain a tip resistance ranging from 0.3 to 2 MΩ. Whole-cell currents of an oocyte were recorded at RT in an extracellular solution with composition specified in figure legends. Electric currents were recorded with a Geneclamp 500B amplifier and Digidata 1322 A AD/DA converter (Molecular Devices, Union City, CA). Currents and voltage were recorded at 200 μs/sample and Bessel filtered at 2 kHz. The data was acquainted and analyzed with software pClamp version 9 (Axon Instruments, Union City, CA). SigmaPlot version 13 (Systat Software, San Jose, CA) and GraphPad Prism version 7 (GraphPad Software, San Diego, CA) were employed for data plotting.

**Surface protein biotinylation**. Biotinylation assays were performed in *Xenopus* oocytes as described[63]. In brief, oocytes were first washed three times using ice-cold PBS solution and then incubated with 0.5 mg/ml sulfo-NHS-SS-Biotin (Pierce, Rockford, IL) at RT for 0.5 h. Non-reacted biotin was quenched with 1 M NH₄Cl. Oocytes were next harvested in CelLytic M lysis buffer (Sigma, St. Louis, MO) plus proteinase inhibitor mixture (Thermo Scientific, Waltham, MA). Lysates were then added with 100 μl 50% streptavidin beads (Pierce) for incubation overnight at 4 °C. The precipitated surface protein was subjected to SDS-PAGE and detected with specific antibodies. Uncropped images of the original scans of immunoblots are shown in Supplementary Figures 12 and 13.

**Immunofluorescence**. Whole mount immunofluorescence experiments in oocytes were carried out as previous[63]. Briefly, oocytes were fixed in 4% paraformaldehyde for 15 min, followed by permeabilization using 0.1% Triton X-100 for 4 min. Oocytes were blocked in 3% skim milk at RT for 30 min, followed by overnight incubation at 4 °C with anti-Flag (#14793, 1:200) or -HA (#3724, 1:200) primary antibodies from Cell Signaling Technology (Danvers, MA), and then incubated with a secondary donkey anti-rabbit IgG conjugated with AlexaFluor 488 (Jackson ImmunoResearch Laboratories, West Grove, PA) for 30 min at RT. Oocytes were mounted in Vectashield (Vector Labs, Burlington, ON) for fluorescence examination using an AIVI spinning disc confocal microscopy (Cell Imaging Facility, Faculty of Medicine and Dentistry, University of Alberta).

**Zebrafish experiments**. Zebrafish experiments were performed as previously described[34]. In brief, embryos of wild-type zebrafish AB strain were kept in E3 solution. PKD2 was knocked down in zebrafish by injection of a translation-blocking antisense morpholino oligonucleotide (MO) (Gene Tools LLC, Philomath, OR) into fertilized eggs within 1 h postfertilization (at 2.5 ng each). The PKD2 MO sequence was: 5′-AGGACGAACGCGACTGGAGCTCATC-3′. Capped mRNAs of human PKD2 WT or mutants were in vitro transcribed with mMessage mMachine T7 kit (Ambion) and co-injected into fertilized eggs with PKD2 MO (at 100 pg each). This study has been approved by the Ethical Committee for Animal Experiments (University of Alberta) and was performed according to the Guidelines for Research with Experimental Animals (University of Alberta) and for the Care and Use of Laboratory Animals (NIH guide, revised in 1996).

**Expression and purification of PKD2 F604P mutant**. The maltose binding protein fused human PKD2 protein, encompassing residues 53–792 and harboring the F604P GOF mutation[5], was expressed and purified from HEK293S GnTI⁻ (ATCC CRL-3022) cells as previously described[9]. In brief, HEK293S GnTI⁻ cells were grown in suspension at 37 °C in freestyle 293 expression medium (Invitrogen, Carlsbad, CA) to the density of ~2 × 10⁶/ml with orbital shaking and then transduced with baculovirus. To boost protein expression, the cell culture was supplemented with sodium butyrate (final concentration of 5 mM) 8–24 h following transduction and maintained at 30 °C. Seveenty-two hours of post-transduction, cells were harvested to obtain crude membrane for affinity purification using amylose resin (New England BioLabs, Ipswich, MA). PKD2 protein was then eluted using buffer containing (in mM) 50 HEPES (pH 7.4), 150 NaCl, 2 TCEP, 0.5 DDM, 20 maltose, and 0.1 mg/ml soybean lipids. PKD2 channels were reconstituted into amphipols A8-35 (Anatrace) as previously described[37]. In brief, the purified PKD2 sample was mixed with amphipols A8-35 at 1:3 (w/w) for 4 h and the detergent was removed with Bio-Beads SM-2 which was then removed with a disposable polyprep column. Next, the channel/amphipols mixture was separated with a Superdex 200 column and collected for cryo-EM analysis.

**Cryo-EM data acquisition**. Purified PKD2 protein sample (3.5 μl, at ~1 mg/ml) was applied to glow-discharged Quantifoil 1.2/1.3 holey carbon 400 mesh copper grids, which were then plunge frozen in liquid ethane with a Vitrobot Mark III (FEI) set at 4 °C, 75% relative humidity, 20 s wait time, -1 mm offset, and 7 s blotting time. Data were collected using a Tecnai TF30 Polara (FEI) operating at 300 kV equipped with a K2 Summit direct electron detector (University of California at San Francisco). Images were then recorded with UCSFImage4[64], using a defocus range between −0.6 and −2.4 μm. Specifically, movies were recorded in a super-resolution counting mode at 31,000× magnification, corresponding to a physical pixel size of 1.2156 Å. The data were collected at a dose rate of 1 e⁻/Å²/frame with a total exposure of 80 frames, giving a total dose of 80 e⁻/Å².

**Image processing, 3D reconstruction and model building**. Movie frames were aligned, dose weighted, and then summed into a single micrograph with MotionCor2[65]. CTF parameters for micrographs were obtained using CTFFIND4[66]. Semi-automated particle selection was performed using Simplified Application Managing Utility for EM Labs (SAMUEL, written by Maofu Liao, Havard Medical School) scripts and displayed in SamViewer (program written by Maofu Liao) for interactively removing ice and protein aggregates. RELION[67] was used for all 2D and 3D processing steps. Particles were sorted into 2D classes with 'bad' particles rejected (i.e., those that are incoherent or poorly resolved classes) by downstream analyses. Specifically, 661,905 particles were initially selected from 1992 micrographs, which was followed by a round of 2D classification and another round of 3D classification to reject bad particles from further analyses (e.g., in silico purification), which lead to a final dataset of ~387,454 particles for subsequent 3D reconstruction. The PKD2 structure (EMD-8354, low-pass filtered to 60 Å), with C4 symmetry imposed, was used as the starting model for 3D classification and refinement. We also carried out reconstruction without imposing 4-fold symmetry, which yielded a map that exhibits C4 symmetry except at disordered loops and the amphipol belt that surrounds the transmembrane region of the channel. Since the map calculated with C4 symmetry was better resolved, we used this map for model building and structural analyses. RELION post-processing using unfiltered half maps with automatic mask resulted in a 3.5 Å resolution map. The automask option in RELION generated a mask that extends 6 pixels beyond a preset density threshold of 0.016. Local resolution was calculated by RELION. UCSF Chimera was used to visualize and segment density maps, and generate figures[68]. HOLE was used to calculate pore radii[69].

The map was sharpened with a b factor of −150 Å² for model building in Coot[70]. The model was refined in real space with PHENIX, and assessed in Molprobity (Supplementary Table 1). To test for overfitting, the model was randomly displaced by 0.2 Å and refined against one of the half maps (half map 1) generated in RELION. FSC curves were then calculated between the refined model vs. both half maps generated in RELION, which revealed a general agreement between the two plots that indicates that the model is not overfitted (Supplementary Figure 4).

**Statistical analysis**. Data were expressed as mean ± SEM (standard error of the mean). No randomization was used to allocate animals to particular groups. The investigators were not blinded to the experimental groups. The number of animals to assess statistical significance of a pre-specified effect was estimated based on our prior experience with the models employed.

**Data availability**. Data supporting the findings of this manuscript are available from the corresponding authors upon reasonable request. The cyro-EM maps of the PKD2 F604P structure have been deposited in the Electron Microscopy Data Bank (EMDB) with the accession code EMD-7786. The atomic coordinate for the corresponding model has been deposited in the Protein Data Bank (PDB) with the accession code 6D1W.

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

## Acknowledgements

This work was supported by the Natural Sciences and Engineering Research Council of Canada (NSERC; to X.-Z.C.), the National Natural Science Foundation of China (grant# 81570648, to X.-Z.C.; grant# 81602448, to J.T.), the Deutsche Forschungsgemeinschaft (DFG, Sonderforschungsbereich 894 and Transregio 152, A14 and P01 to V.F.) and the National Institutes of Health (NIH; Grant DK102092, to Y.Y.; R01 DK110575-01 to E. C.). E.C. is a Pew Scholar. W.Z. was a recipient of the Alberta Innovates-Doctoral Graduate Student Scholarship. R.C. and Q.H. were recipients of the NSERC IRTG Studentship. L.H. was a recipient of the International Research Training Group 1830 Student Scholarship (DFG). We thank the Utah Center for High Performance Computing for computational support and David Belnap for data collection at the electron microscope core at the University of Utah. We thank Yifan Cheng for help with data collection at UCSF and Peter Shen for helpful discussion and advice.

## Author contributions

Conceptualization, W.Z., X.Y., Y.Y., J.T., V.F., Y.C., E.C., X.-Z.C.; Investigation, W.Z., X.Y., R.H., R.C., L.H., Z.W., Q.H., X.L., D.B.; Writing, W.Z., Y.Y., E.C., X.-Z.C.

## Additional information

**Competing interests:** The authors declare no competing interests.

