## [Peer Review File · Nature Communications]

Reviewers' comments:

Reviewer #1 (Remarks to the Author):

The authors present data to show that they have identified a two residue gate in PKD2L1 and a one residue gate in PKD2 (L677). They showed extensive electrophysiology data on a series of mutant proteins for PKD2L1 and additional data on PKD2. These mutations involve the residues in the previously identified lower gate in PKD2. They showed that the PKD2 activatory mutation L677G rescues tail curling and cyst formation in zebrafish embryos caused by knocking down PKD2. They also present the structure of PKD2 with a different activating mutation, F604P at 3.5Å by cryo-EM. This mutation on S5 causes the central kinked pi-helix of S6, adjacent to the S5 F604P mutation, to convert to a straight S6 alpha helical arrangement, which moves the distal end of S6, such that the L677 residue moves, and the lower gate is opened.

In general this is an excellent paper with very high quality data, which provides substantial new insights into the function of the PKD channels. The conclusions they draw from the data are generally sound and the presentation is of high quality. I am not aware of any similar results having been published or presented elsewhere, so to the best of my knowledge I believe that the results are novel.

One problem with this paper is that the use of English is not good, there are numerous errors in the language which unfortunately would make it difficult to follow for someone who does not know the field. The paper will need extensive copy editing to correct these errors. I have provided a list of possible changes to improve the English in a "minor corrections" section below, but this is in no way exhaustive. The limitations in the English do not however detract from the excellent quality of the science described in the paper.

In the abstract the authors imply that the effects they observe are significant in disease, although this is of course somewhat difficult to prove. Still, it is the first mechanism for the opening of a PKD channel, so it is very valuable to have this information.

Introduction: Are the mutations they discuss actually found in disease? They are talking about activating gating mutations. Are these the cause of disease, or do they simply mean that a general understanding of gating might help explain disease mutations, or help to show activation.

In general throughout the paper it would be useful for the authors to be clear when they are talking about disease related mutations and when they are discussing site-directed mutagenesis changes made to study channel function, but not seen in patients or the general population.

It would be very useful if they indicated clearly in their numbering when they are referring to PKD2L1 and when they are talking about PKD2, perhaps with an indication of the numbering for PKD2. It is confusing to talk about L557 in PKD2L1 and the same residue, L677 in PKD2, without indicating that these two residue are the same in an alignment. They could add the protein name in superscript, or select the numbering for one protein and use that, or refer to the other protein numbering in brackets.

Page 5: Inspection of the available PKD2 structures raises the question as to why the A558 mutations have the same effect as the L557 mutation, as L557 points into the central cavity, whereas A558 points away from the central cavity. This is not very clearly explained in the following section.

Page 5 and Figure 2B: The author state "Mutations at site A558 revealed similar results (Fig. 2B), with the exception that A558T and A558G did not exhibit any current increase at -50 mV compared to the WT channel (Fig. 2B)." This reviewer is not convinced that the results are "similar". There are 4 residues out of 13 that are significantly different, so that seems rather

dissimilar to me. Nor would I expect the results to be similar, given the positions of the two residues. I suggest they phrase this paragraph differently.

Page 7 "However, because of its hydrophilicity it should be less likely to form a pore gate." It is less likely to form a hydrophobic gate, but why should it not form a gate, albeit not a hydrophobic gate?

Page 7: "the GOF of isosteric mutant L677N suggested that the constriction formed by N681 does not effectively block ion flow".

Perhaps the L677N mutation causes N681 to move, preventing it from gating the channel. Two Asn residues could interact with each other, moving N681 and preventing it from gating the pore. In general the interpretations in this section might be true, but they have not been proved conclusively. I think toning down the language would be helpful in this part.

The authors refer to there being two gating residues for PKD2L1 and one PKD2. Although their data does tend to support this idea, it is possible that other gating mechanisms might use only one in PKD2L1 and/or two in PKD2, or indeed might not involve this type of change of conformation at all. The authors should be careful to always indicate that this is one of many potential gating mechanisms for the PKD2 channels.

One obvious criticism of this work is that, due to the difficulty of measuring a basal current for WT PKD2, the authors were forced to do some of their studies in PKD2 and some in PKD2L1. Ideally all the results would be duplicated in both proteins before they make statements that are relevant to both proteins, but in practice this is not practical and they have been very successful in bringing together a great deal of information on both channels. Taken together this data does provide a very strong story, even though it was not possible to do the complete study with both channels.

It seems likely that PKD2 could be opened by a number of diverse changes in the structure. The two mutations described in this paper (604 and 677) do both open the channel effectively, which is a very nice result. Whether these particular changes in the structure are important in normal channel gating in cells is not clear. It is of course very difficult to determine how exactly gating occurs in normal cells, without a physiological agonist. Never-the-less these results are a very useful and interesting contribution to our understanding of the function of this channel.

In the opinion of this reviewer it is definitely worth publishing this paper. As far as I can tell the data is strong, the story is consistent and the results provide a clear and very useful interpretation of one gating mechanism for PKD2 and PKD2L1.

The work will definitely require extensive copy editing to correct the many errors in the language, but this in no way detracts from the high quality of the data and the highly significant results.

Minor comments

As mentioned above the paper has many grammatical errors, due to the arcane nature of the English language. I have commented on a number of the places where the language could be improved, in an effort to help the authors. I hope that they will not take this as criticism, it is intended to be of assistance. There are many other places where the language could be improved and I hope that the journal editors will assist the authors with corrections to these minor errors.

Pg 3 , second paragraph "PKD2 shares overall structures with other TRP channels" isn't really clear, should be something like "PKD2 has a similar overall fold to other TRP Channels"

PKD1 and PKD2 have been shown to form a complex in "the" primary cilia of renal tubule epithelial cells, and "they are" proposed to act as a fluid flow sensor, although "this" hypothesis was challenged recently.

"still regarded as untreatable"

"with marginal, if not debatable," – I am not clear what the authors mean here, do they mean marginal, if any effects, or that the quality of the data is debatable, so they results should be disregarded.

Page 4 "recently resolved high resolution of PKD2 structure" should perhaps say "recently resolved high resolution PKD2 structures"

Page 4 "it remains still elusive whether both L677 and N681 residues or just single residue acts as a functional gate in PKD2" should perhaps say "it remains unclear whether both L677 and N681 residues or just one of these residues acts as a functional gate in PKD2".

Page 4 "However, ligand induced open state structure" should perhaps read ""However, a ligand induced open state structure"

Page 4 "we obtained high resolution structure of PKD2" should say "we obtained a high resolution structure of PKD2"

Page 5 when talking about mutants eg L557N or A558N, it should read "expression of the L557N or A558N mutant proteins" the L447N is just the mutation, not the whole expressed protein.

Page 5 "we found that the plasma membrane expression levels of these mutants"

Page 5: amino acid residues, not just amino acids

Page 5: "is in strong agreement with" would be better just as "is in agreement with"

Line 4 of Page 7: "an activated state conformation" – should this be "activated state" or "activated conformation"?

Line 5 on Page 7: "we found that F604P function" – do you mean F604P activation

End of page 7: "to a more pronouncing rescuing effect" should be pronounced not pronouncing. There is a similar error at the top of page 8.

Top of page 8: "Because the agonist of PKD2 homotetrameric channel is unknown, to elucidate the activation mechanism of PKD2 channel we resolved structure of PKD2 F604P GOF mutant" is not very clear, perhaps use instead "Because there is no known agonist for the PKD2 homotetrameric channel, to elucidate the activation mechanism of the PKD2 channel we solved the structure of PKD2 F604P GOF mutant". This might make more sense.

"We determined a cryo-EM structure of the ..."

"using a truncated hPKD2 construct..."

"in a kink in S5"

"became disordered and bulged outward" might be better.

Page 14: "PKD2 was knockdown in zebrafish" – should this be "knocked down"

Page 15: "then reconstituted into amphipols" – should be "reconstituted"

At this point this reviewer gave up on editing the text. There are numerous additional errors, which no doubt the editors will fix during the editing process. These linguistic errors make the paper difficult to read, but they do not detract from the value of the work, which is definitely worth publishing in Nat Comms in the opinion of this reviewer.

Reviewer #2 (Remarks to the Author):

Chen and colleagues present a study of gating mechanisms in PKD2 and PKD2L1 channels, which is composed of two parts: 1. Mutagenesis, electrophysiology, and in vivo assays that demonstrate PKD2 and PKD2L2 channels operate by the "hydrophobic gating mechanism," and 2. A cryo-EM structure of a PKD2 F604P GoF mutant suggesting a pi-helix to alpha-helix transition in S6 during opening.

The mutant PKD2 structure that is presented is interesting and could have potential to provide novel insight. However, the manuscript in its current form is not suitable for publication in Nature Communications without addressing two critical issues by performing additional experiments.

Major comments

1. Cryo-EM structure of PKD2 F604P and its implication on the gating mechanism

Quality of the cryo-EM data and model building.

The reconstruction shown in Fig. S3B does not contain structural details that one would typically observe in a 3.5 to 4 Å resolution structure. No side chain densities are clearly visible, and it is unclear how the authors were able to build a reliable model into the density. No close-up views showing high quality cryo-EM density with well-resolved side chains are shown, calling into question the validity of the atomic model. Side chains should be clearly visible at 3.5 Å resolution. Based on the quality of the reconstruction shown in Fig. S3B, the density appears to be largely in the 4-6 Å resolution range, unless the reconstruction shown in Fig. S3B is an unsharpened map. It is possible that the particles exhibited conformational heterogeneity and that C4 symmetry was forced, which would result in inflated resolution without the structural features that are consistent with the reported resolution. This would also explain the very high B-factor ($\sim 180 \text{Å}^2$) that was applied to the data for model building. Asymmetric 3D classification and reconstruction should be performed in order to ensure that the particles are truly C4 symmetric and that the mutation did not perturb the structural integrity of the complex. Also, there was no mention in the methods of the type of mask that was used to improve the map from 3.8 Å resolution, as is reported in the methods, to 3.5 Å, for the final structure described throughout the text. This mask should be included in a reconstruction flow chart.

In addition, the following important supplemental materials are missing:

- Close-up views of the cryo-EM density map for each structural segment (S1-S6, ToP domain), especially the region around S5 and S6 where the pi to alpha helical transition occurred.
- The cryo-EM data statistic table is missing – there are no modelling statistics at all.
- Model-map FSC plot, including half maps.
- 2D class averages.
- A 3D reconstruction flow chart, including asymmetric reconstruction.
- A local resolution plot.
- Euler distribution plots.

The mechanism of F604P.

- Is it possible that the F604P mutant structure captured is not in an open state, but is rather in a desensitised or inactive state? The F676/I680 residues could, in this instance, be forming a hydrophobic cuff that would prevent ion conduction. Based on the hydrophobic gating theory that was put forth by the authors, these residues should provide a significant energetic barrier. Also, by thermodynamic reasoning, wouldn't it be more likely that the high-energy pi helix is present in an

open state, rather than alpha helix?

- The authors proposed that the selectivity filter of PKD2 acts as a gate, like in TRPV1. If this is in fact the case, the active mutant PKD2 F504P should adopt the open state. However, it appears that the selectivity filter of F504P is in a nonconductive state, very similar to wild-type PKD2. The upper gate (selectivity filter) in TRPV1 clearly acts as an activation gate, based on the comparison between the apo and RTx/DkTx-bound activated states. However, in the PKD2 structure, the selectivity filter adopts non-conducting states. Observing a non-conductive state of the filter could mean the inactive state or simply observed a state that is off the pathway, not necessary suggesting that it will act as a gate. These points should be addressed.
- I believe the idea that the pi-helix in the S6 activation gate acts as a hinge, which allows splaying of the helix bundle gate and can also transition to an alpha-helix, was first suggested based on comparison of the TRPV2 and TRPV1 channels (Zubcevic et al, NSMB, 2016). The authors should be transparent and clearly acknowledge the origin of this idea in the text.

2. Hydrophobic gating and mutagenesis studies.

The term "hydrophobic gating" relates mainly to channels that do not possess a classic helix bundle crossing gate, and have a large diameter across the lower pore, which makes them appear to be permeable to ions (i.e. they lack residues that create a physical constriction in the pore). But because of the hydrophobic nature of their pores, the energy barrier for permeation of hydrated ions is so large that their pores are effectively closed. This mechanism, in its pure form, does not seem to apply to the PKD2 channels because the residues that act as a designated gate form a tight constriction within the helix bundle crossing in the PKD structure. While it is true that the residues forming the helix bundle crossing are hydrophobic, the channels are not only gated by the principle of hydrophobicity, but also by a physical opening of the activation gate (gating – wider or narrower opening). Therefore, the premise for the experimental setting is flawed; introduction of hydrophilic residues at the hydrophobic gate will increase conductance if the channel works by the hydrophobic gating mechanism. The problems with the experimental setting are that mutation can influence open probability (gating) as well as ion conductance (permeation). The authors' hydrophobic gating infers that the mutation does not affect gating. This is not the case. The Swartz group performed elegant studies using the Shaker K⁺ channel demonstrating that the mutation of hydrophobic residues at the gate affect the channel gating and their effect on the conductance is minimal (Kitaguchi et al, 2003, Sukhareva M et al, 2003). Along the same line, I believe that the Tucker group introduced a charged residue into the pore-lining S6 and obtained the open structure of KirBac 3.1 (Bavro et al, NSMB 2012). Because of this reason, the authors' interpretation of the PKD2L1 and PKD2 mutagenesis data in the context of purely hydrophobic gating is not convincing.

- To maintain their claim of hydrophobic gating, the authors must show that the mutations do not affect the single channel conductance but not gating either by single channel measurement or noise analysis or dose-response curve (if ligand or activating stimuli is known), like the Swartz group did with the Shaker K⁺ channel. If authors cannot provide the relevant functional data due to the technical difficulty, authors should determine the structure of PKD2 L677N mutant to test if there is any physical widening of the region at N677 as compared to wild type.
- In my view the authors' studies are simply to identify critical residues that form a gate. In PKD2L1, substitutions of the gate residue L557 with a polar residue of an equivalent length would influence the gating (pore open wide) rather than the simple change in hydrophobicity. This is actually evident in the mutagenesis data presented; substitutions of Leu577 with longer polar residues cause an increase in channel activity and substitutions with smaller polar residues (S, C, T) give similar results as smaller hydrophobic residues (A, G). This suggests that the increased channel activity may stem from changes in effective pore diameter rather than from decreasing hydrophobicity of the pore, even without invoking the change in gating. Also, substitution of Leu to bulky Trp results in the nearly complete loss of the currents. Trp is not the most hydrophobic amino acid, but it is the largest amino acid. Notably, the Swartz group also observed the Trp mutation caused the channel to prefer the closed state but had little effect on the conductance.
- It looks like that the PKD construct that was used for functional studies is full-length, not the truncated construct that was used for cryo-EM studies. The authors should provide data showing

that the construct that was used for cryo-EM (53-792) is functional and that functional phenotype remains unchanged when mutated (F604P).

- The authors use the term "hydrophobic seal" distinctly from "physical steric occlusion" of the pore without much explanation. This is quite confusing and incorrect. "Hydrophobic seal" was originally used to describe the hydrophobic core of the voltage-sensor that physically occludes ion permeation through the voltage sensor. Also, the Tucker and Sansome's original paper of "hydrophobic gating" did not use "hydrophobic seal." In several places, I think authors interchange "hydrophobic seal" and "physical steric occlusion." The authors should define something like a "hydrophobic barrier," in which the opening is large enough to permeate ions, but due to its hydrophobicity cannot.

Minor points:

- Pi-helices in S6 have been observed in other WT PKD channels and the closely related TRPML channels. It would be useful to have a comparison of the pores in related channels with gate/restricting residues annotated (i.e. a modified version of figure S5 (S6 in legends)) along with a discussion on the questions posed above.
- p.3, 5th line from bottom: "activated mutant" should be "activating mutant"
- p. 4, 6th line from top. The sentence is somewhat unclear; it needs a rewrite.
- Order of Fig. S4-S6, along with figure legends, is wrong.
- Figure S5 (S6 in legends) is not very informative (the pi helix and alpha helix in TRP channels). Align the structures so that the same view is obtained for all channels. It would also be helpful to show gate residues for each of the channels. Also, it'd be nice to see both open and closed conformations for channels where both are available (i.e. TRPV1).
- The in vivo complementation assays utilize L677G, which is not the most potent GoF mutant of the PKD2 channel. It does not appear to be explained in the text why this mutant was preferred over L677N. In addition, complementation with WT PKD2 does not prevent formation of pronephric cysts; is there a reason for this? The in vivo assay does not add much value to the study. I would recommend removing this section from the MS and moving the corresponding figure to supplementary information.

Reviewer #3 (Remarks to the Author):

This study examined the gating of PKD2 and PKD2L1 and suggested a novel hydrophobic gating mechanism which provided insights into how these channel function. A strength of this work is that multiple techniques were employed to examine the gating of these channel proteins. Two-electrode voltage clamp in frog oocytes was used to identify critical residues that would lead to gain-of-function (GOF) or loss-of-function (LOF) mutations in the proteins. They then expressed these GOF mutant proteins to rescue PKD2 knockdown zebrafish, where (the KD fish were sick without the GOF PKD2. They then obtained high resolution structures of PKD2 to untangle the changes in the protein that lead to the altered gating. The authors have developed a clever model that suggests that the closing the pore occurs through forming a hydrophobic seal. A combined twisting and splaying motion of gate residues during PKD2 activation leads to opening of the hydrophobic gates. Moreover, they show that PKD2L1 utilizes two residues for the hydrophobic gate and PKD2 utilizes a single residue.

There are limitations to this study. One limitation is that it is assumed that the reader will know that the full length protein is expressed in cells and fish to get functional results and the truncated protein is utilized to examine the structural changes. This difference needs to be explicitly stated. Also, the limitations of comparing function using a full protein and structure using a truncated protein needs to be discussed.

Another limitation is the lack of acknowledgment that calcium is an agonist of PKD2 channel activity and PKD2 dependent currents. The statement (on page 3) "the agonist for PKD2 homotetrameric channel has not been identified" is misleading. It is true that it would be ideal to

identify an agonist other than calcium. However, there are many publications showing that the PKD2 channel is activated by cytoplasmic calcium. Both single channel measurements and calcium release from intracellular stores are activated by increases in cytoplasmic calcium. The effects of calcium as an agonist must be addressed here and throughout the manuscript. For example, at the bottom of page 4, the authors say that the ligand-induced open state structure cannot be resolved until an agonist is identified. But the effects of calcium are not considered. And at the beginning of the Allosteric section of Results on page 8 it is stated that the agonist of PKD2 is unknown without mentioning calcium as a possible agonist. Of course, the authors may want to ignore calcium as an agonist because the calcium binding sites that may confer ligand-induced openings are in the cytoplasmic domains, regions that are not included in the structural analysis. This is a problem of the study, and it must be addressed in this manuscript.

In the introduction, the authors claim that the hypothesis that PKD1 and PKD2 act as a fluid flow sensor has been challenged and they cite the work of only one group. This is not correct as a number of groups have experimental results that question this hypothesis. At the same time, several groups have strong evidence to support the hypothesis. The relevance of cilia to clinical outcomes is also unclear. If these ideas are going to be discussed, a more balanced approach is needed.

The statement "Compared with PKD2, its homologue PKD2L1 (also called polycystin-2L1 or TRPP3), which is not involved in ADPKD, has been more extensively studied as an ion channel" is not correct. The authors need to be certain that their statements such as this can be supported unequivocally.

On page 5, in the statement "Compared to the general functional characteristics of PKD2 and PKD2L1 channels, it has remained more elusive", it is not clear what has remained elusive.

Bottom of the first paragraph on page 5, is it correct to refer to "L577 and A558" or should it be L557 and A558? Also in Fig 1 panel B the currents are labeled 577 and 578, but in other panels of Fig 1 it is 557 and 558. This is very confusing. Please check and correct. In fact, please check all residue numbers throughout the manuscript.

Bottom of page 7. "pronouncing" should be "pronounced". And "observed larger channel activity" should be "observed larger current" – channel activity is not measured, ion currents are measured here.

The effect shown in Fig 4 D is claimed to be a "significant rescue" but the effect is only 50%. An explanation of why the effect is not more complete is needed.

The size of the PKD2 pore has been functionally assessed previously (Anyatonwu JBC 2005). This work needs to be included in the comparison.

On page 11, there is speculation about the different effects that would be evoked by one or two molecules of calcium binding. The effect of the number and location of EF hand motifs in the cytoplasmic domain of PKD2 was studied and found to regulate the calcium dependence of PKD2 function (Kuo, FASEB J 2014). Discussion of how calcium binding in the cytoplasmic domain needs to be included in the manuscript.

Responses to reviewers' comments (bold and start with ">>"; others are referees' original comments)

Reviewer #1 (Remarks to the Author):

The authors present data to show that they have identified a two residue gate in PKD2L1 and a one residue gate in PKD2 (L677). They showed extensive electrophysiology data on a series of mutant proteins for PKD2L1 and additional data on PKD2. These mutations involve the residues in the previously identified lower gate in PKD2. They showed that the PKD2 activatory mutation L677G rescues tail curling and cyst formation in zebrafish embryos caused by knocking down PKD2. They also present the structure of PKD2 with a different activating mutation, F604P at 3.5A by cryo-EM. This mutation on S5 causes the central kinked pi-helix of S6, adjacent to the S5 F604P mutation, to convert to a straight S6 alpha helical arrangement, which moves the distal end of S6, such that the L677 residue moves, and the lower gate is opened.

In general this is an excellent paper with very high quality data, which provides substantial new insights into the function of the PKD channels. The conclusions they draw from the data are generally sound and the presentation is of high quality. I am not aware of any similar results having been published or presented elsewhere, so to the best of my knowledge I believe that the results are novel.

One problem with this paper is that the use of English is not good, there are numerous errors in the language which unfortunately would make it difficult to follow for someone who does not know the field. The paper will need extensive copy editing to correct these errors. I have provided a list of possible changes to improve the English in a minor corrections section below, but this is in no way exhaustive. The limitations in the English do not however detract from the excellent quality of the science described in the paper.

>> We are very grateful to the reviewer for the very positive appraisals of our study and we appreciate the reviewer's efforts in correcting linguistic errors present in this manuscript. We have now polished the manuscript in terms of grammar, spelling and style, including making corrections suggested by the reviewer.

In the abstract the authors imply that the effects they observe are significant in disease, although this is of course somewhat difficult to prove. Still, it is the first mechanism for the opening of a PKD channel, so it is very valuable to have this information.

Introduction: Are the mutations they discuss actually found in disease? They are talking about activating gating mutations. Are these the cause of disease, or do they simply mean that a general understanding of gating might help explain disease mutations, or help to show activation.

>> The gain-of-function mutation F604P identified in a previous study (Arif Pavel et al, PNAS, 2016) and gain-of-function and loss-of-function gate mutations identified in this study are not naturally occurred pathogenic/disease mutations found in human patients. They are used to explore the mechanisms underlying the PKD2 channel function and gating, which is of fundamental importance to understanding the function of disease mutants and for therapeutic interventions.

In general throughout the paper it would be useful for the authors to be clear when they are talking about disease related mutations and when they are discussing site-directed mutagenesis changes made to study channel function, but not seen in patients or the general population.

>> We have now specified that F604P is a mutation which was used as a tool to study PKD2 channel function (page 3, lines 5-7 from the bottom). For human disease mutations, we always specified accordingly, e.g., page 7, lines 4-5 from bottom.

It would be very useful if they indicated clearly in their numbering when they are referring to PKD2L1 and when they are talking about PKD2, perhaps with an indication of the numbering for PKD2. It is confusing to talk about L557 in PKD2L1 and the same residue, L677 in PKD2, without indicating that these two residue are the same in an alignment. They could add the protein name in superscript, or select the numbering for one protein and use that, or refer to the other protein numbering in brackets.

>> Excellent suggestion. We have now provided descriptions to relate PKD2L1 L557 to PKD2 L677 (page 5, line 10; page 7, line 5).

Page 5: Inspection of the available PKD2 structures raises the question as to why the A558 mutations have the same effect as the L557 mutation, as L557 points into the central cavity, whereas A558 points away from the central cavity. This is not very clearly explained in the following section.

>> Excellent point. A likely reason is that PKD2L1 might possess a slightly different ion-conducting pore from PKD2, i.e., PKD2 contains a single-residue gate in which L677 directly points to the central cavity (with A678 pointing away from the central cavity) whereas PKD2L1 contains a two-residue gate in which both L557 and A558, according to our functional data, would face the central cavity but neither of them would directly point to the cavity centre (see illustration below). Therefore, mutations at either PKD2L1 L557 or A558 would alter the local hydrophobicity and thus the channel activity. We have now added this comparison in Results (page 6, lines 5-10).

Page 5 and Figure 2B: The author state Mutations at site A558 revealed similar results (Fig. 2B), with the exception that A558T and A558G did not exhibit any current increase at -50 mV compared to the WT channel (Fig. 2B). This reviewer is not convinced that the results are similar. There are 4 residues out of 13 that are significantly different, so that seems rather

dissimilar to me. Nor would I expect the results to be similar, given the positions of the two residues. I suggest they phrase this paragraph differently.

>> We agree with the reviewer's comment and have now rephrased as "Mutations at A558 in general also revealed a positive correlation between the hydrophilicity of site 558 and channel activity (Fig. 2b), with the exception that.....".

Page 7 However, because of its hydrophilicity it should be less likely to form a pore gate. It is less likely to form a hydrophobic gate, but why should it not form a gate, albeit not a hydrophobic gate?

>> Agreed. We have now changed it to "However, because of its hydrophilicity we think that N681 is not as effective as L677 to block ion flow".

Page 7: the GOF of isosteric mutant L677N suggested that the constriction formed by N681 does not effectively block ion flow. Perhaps the L677N mutation causes N681 to move, preventing it from gating the channel. Two Asn residues could interact with each other, moving N681 and preventing it from gating the pore. In general the interpretations in this section might be true, but they have not been proved conclusively. I think toning down the language would be helpful in this part.

>> We agree to apply cautiousness and have now softened the tone (page 8, lines 3-6).

The authors refer to there being two gating residues for PKD2L1 and one PKD2. Although their data does tend to support this idea, it is possible that other gating mechanisms might use only one in PKD2L1 and/or two in PKD2, or indeed might not involve this type of change of conformation at all. The authors should be careful to always indicate that this is one of many potential gating mechanisms for the PKD2 channels.

>> We thank the reviewer for raising this important point. We have now specified that this is one of many potential gating mechanisms for PKD2 channels in several places of the manuscript (e.g., page 10, lines 1-3 from the bottom; page 13, lines 9-11).

One obvious criticism of this work is that, due to the difficulty of measuring a basal current for WT PKD2, the authors were forced to do some of their studies in PKD2 and some in PKD2L1. Ideally all the results would be duplicated in both proteins before they make statements that are relevant to both proteins, but in practice this is not practical and they have been very successful in bringing together a great deal of information on both channels. Taken together this data does provide a very strong story, even though it was not possible to do the complete study with both channels.

>> We thank the reviewer for the understanding and recognition.

It seems likely that PKD2 could be opened by a number of diverse changes in the structure. The two mutations described in this paper (604 and 677) do both open the channel effectively, which is a very nice result. Whether these particular changes in the structure are important in normal channel gating in cells is not clear. It is of course very difficult to determine how exactly gating occurs in normal cells, without a physiological agonist. Nevertheless these results are a very useful and interesting contribution to our understanding of the function of this channel.

>> To echo and incorporate the reviewer's thoughts we have now described the known ways of opening PKD2 channels, including the effects of Ca²⁺, in Discussion (page 13, second paragraph).

In the opinion of this reviewer it is definitely worth publishing this paper. As far as I can tell the data is strong, the story is consistent and the results provide a clear and very useful interpretation of one gating mechanism for PKD2 and PKD2L1.

The work will definitely require extensive copy editing to correct the many errors in the language, but this in no way detracts from the high quality of the data and the highly significant results.

>> We are thankful to the reviewer's enthusiasm and recognition.

Minor comments

As mentioned above the paper has many grammatical errors, due to the arcane nature of the English language. I have commented on a number of the places where the language could be improved, in an effort to help the authors. I hope that they will not take this as criticism, it is intended to be of assistance. There are many other places where the language could be improved and I hope that the journal editors will assist the authors with corrections to these minor errors.

>> We are grateful to the reviewer's efforts on the editing.

Pg 3 , second paragraph PKD2 shares overall structures with other TRP channels isn't really clear, should be something like PKD2 has a similar overall fold to other TRP Channels

>> The sentence has now been changed to "PKD2 shares an overall architecture with other TRP channels".

PKD1 and PKD2 have been shown to form a complex in the primary cilia of renal tubule epithelial cells, and they are proposed to act as a fluid flow sensor, although this hypothesis was challenged recently.

>> Changed.

still regarded as untreatable

>> Changed.

with marginal, if not debatable, – I am not clear what the authors mean here, do they mean marginal, if any effects, or that the quality of the data is debatable, so they results should be disregarded.

>> We have now changed it to "with marginal effectiveness".

Page 4 recently resolved high resolution of PKD2 structure should perhaps say recently resolved high resolution PKD2 structures

>> Changed.

Page 4 it remains still elusive whether both L677 and N681 residues or just single residue acts as a functional gate in PKD2 should perhaps say it remains unclear whether both L677 and N681 residues or just one of these residues acts as a functional gate in PKD2.

Page 4 However, ligand induced open state structure should perhaps read However, a ligand induced open state structure

Page 4 we obtained high resolution structure of PKD2 should say we obtained a high resolution structure of PKD2

>> All Changed.

Page 5 when talking about mutants eg L557N or A558N, it should read expression of the L557N or A558N mutant proteins the L447N is just the mutation, not the whole expressed protein.

>> Changed and applied the principle across the manuscript.

Page 5 we found that the plasma membrane expression levels of these mutants

Page 5: amino acid residues, not just amino acids

Page 5: is in strong agreement with would be better just as is in agreement with

>> All Changed.

Line 4 of Page 7: an activated state conformation – should this be activated state or activated conformation ?

>> Changed to “an activated conformation”.

Line 5 on Page 7: we found that F604P function – do you mean F604P activation

>> We have now changed it to “we found that the channel activity of F604P mutant...”.

End of page 7: to a more pronouncing rescuing effect should be pronounced not pronouncing. There is a similar error at the top of page 8.

>> Both changed.

Top of page 8: Because the agonist of PKD2 homotetrameric channel is unknown, to elucidate the activation mechanism of PKD2 channel we resolved structure of PKD2 F604P GOF mutant is not very clear, perhaps use instead Because there is no known agonist for the PKD2 homotetrameric channel, to elucidate the activation mechanism of the PKD2 channel we solved the structure of PKD2 F604P GOF mutant . This might make more sense.

>> Agreed. We have now made the changes.

We determined a cryo-EM structure of the ...

using a truncated hPKD2 construct...

in a kink in S5

became disordered and bulged outward might be better.

>> All Changed.

Page 14: PKD2 was knockdown in zebrafish – should this be knocked down

>> Changed.

Page 15: then reconstituted into amphipols – should be reconstituted

>> Changed.

At this point this reviewer gave up on editing the text. There are numerous additional errors, which no doubt the editors will fix during the editing process. These linguistic errors make the paper difficult to read, but they do not detract from the value of the work, which is definitely worth publishing in Nat Comms in the opinion of this reviewer.

>> We thank the reviewer for the patience. We believe that the manuscript is now easier to follow.

Reviewer #2 (Remarks to the Author):

Chen and colleagues present a study of gating mechanisms in PKD2 and PKD2L1 channels, which is composed of two parts: 1. Mutagenesis, electrophysiology, and in vivo assays that demonstrate PKD2 and PKD2L2 channels operate by the hydrophobic gating mechanism, and 2. A cryo-EM structure of a PKD2 F604P GoF mutant suggesting a pi-helix to alpha-helix transition in S6 during opening. The mutant PKD2 structure that is presented is interesting and could have potential to provide novel insight. However, the manuscript in its current form is not suitable for publication in Nature Communications without addressing two critical issues by performing additional experiments.

Major comments

1. Cryo-EM structure of PKD2 F604P and its implication on the gating mechanism

Quality of the cryo-EM data and model building.

The reconstruction shown in Fig. S3B does not contain structural details that one would typically observe in a 3.5 to 4 resolution structure. No side chain densities are clearly visible, and it is unclear how the authors were able to build a reliable model into the density. No close-up views showing high quality cryo-EM density with well-resolved side chains are shown, calling into question the validity of the atomic model. Side chains should be clearly visible at 3.5 resolution. Based on the quality of the reconstruction shown in Fig. S3B, the density appears to be largely in the 4-6 resolution range, unless the reconstruction shown in Fig. S3B is an unsharpened map. It is possible that the particles exhibited conformational heterogeneity and that C4 symmetry was forced, which would result in inflated resolution without the structural features that are consistent with the reported resolution. This would also explain the very high B-factor (-1802) that was applied to the data for model building. Asymmetric 3D classification and reconstruction should be performed in order to ensure that the particles are truly C4 symmetric and that the mutation did not perturb the structural integrity of the complex. Also, there was no mention in the methods of the type of mask that was used to improve the map from 3.8 resolution, as is reported in the methods, to 3.5, for the

final structure described throughout the text. This mask should be included in a reconstruction flow chart.

>> We thank the reviewer for raising these points about our cryo-EM data and model building.

Indeed, in the initial submission, only an unsharpened, segmented map was presented. We agree that -180 \AA^2 b factor is a bit high, therefore we have now used -150 \AA^2 b factor for preparing figures in resubmission. A b factor of -150 \AA^2 adequately brings out high resolution features of the map. Also, as suggested, we have performed asymmetric 3D classification and reconstruction, and found that the particles are truly C4 symmetric (Supplementary Figure 7).

Regarding the mask used, in the postprocessing within RELION, there is an automask option. We used this option to generate a mask that extends 6 pixels beyond a preset density threshold of 0.016. This mask is quite generous to include all true channel density, while flattening out surrounding noise. We have now provided this information in Methods (page 17, lines 3-5 from bottom).

In addition, the following important supplemental materials are missing:

- Close-up views of the cryo-EM density map for each structural segment (S1-S6, ToP domain), especially the region around S5 and S6 where the pi to alpha helical transition occurred.

>> These have been now included in Supplementary Figure 6.

- The cryo-EM data statistic table is missing – there are no modelling statistics at all.

>> The modelling statistics have been provided in Supplementary Table 1.

- Model-map FSC plot, including half maps.

>> Included in Supplementary Figure 4.

- 2D class averages.

>> Included in Supplementary Figure 4.

- A 3D reconstruction flow chart, including asymmetric reconstruction.

>> Included in Supplementary Figure 5 and 7.

- A local resolution plot.

>> Included in Supplementary Figure 4.

- Euler distribution plots.

>> Included in Supplementary Figure 5 and 7.

The mechanism of F604P.

•Is it possible that the F604P mutant structure captured is not in an open state, but is rather in a desensitized or inactive state? The F676/I680 residues could, in this instance, be forming a hydrophobic cuff that would prevent ion conduction. Based on the hydrophobic gating theory that was put forth by the authors, these residues should provide a significant energetic barrier. Also, by thermodynamic reasoning, wouldn't it be more likely that the high-energy π helix is present in an open state, rather than α helix?

>> We thank the reviewer for raising this important point.

First of all, comparison of TRPV1 structures in apo and capsaicin-bound activated states revealed that capsaicin induces pore dilation at the hydrophobic lower gate from 5.3 Å to 7.6 Å (Cao et al, *Nature*, 2013). Similarly, chemical agonist ML-SA1 induced pore dilation of TRPML1 from 5.6 Å to 8.4 Å (Schmiege et al, *Nature*, 2017). These observations suggest that the hydrophobic gate would exert an energy barrier when the pore diameter is within 6 Å and the pore would be open if the pore diameter is increased to 7.6 Å. In our study, the PKD2 F604P mutant structure exhibited a pore diameter of 7.6 Å at the lower gate (See Fig. 6b) while our recently reported PKD2 WT structure in a closed state has a pore diameter of 5.0 Å (Shen et al, *Cell*, 2016). All these data together suggest that mutant F604P would unlikely represent a desensitized or inactivated state, although we can't exclude the possibility that it may be in a partially open state and be different from activated states of WT channel induced by its physiological agonists. We have now included this discussion in the revised manuscript (page 10, lines 4-9).

Second, regarding the reviewer's comment that the high-energy π -helix, but not the α -helix, would be present in an open state, in fact, during the revision of this manuscript, two cryo-EM structures of human TRPV6 in open and closed states, respectively, were published and the authors indeed observed a π -helix mode within S6 in the open structure and an α -helix in the closed structure (McGoldrick et al, *Nature*, 2018). However, among all the TRP channels whose structures have been resolved to date, most of them have a π -helix within S6 in closed states, including TRPV1 (Liao et al, *Nature*, 2013), TRPA1 (Paulsen et al, *Nature*, 2015), PKD2 (Shen et al, *Cell*, 2016), TRPML1 (Schmiege et al, *Nature*, 2017), TRPML3 (Hirschi et al, *Nature*, 2017), TRPM4 (Autzen et al, *Science*, 2017; Winkler et al, *Nature*, 2017), and NOMPC (Jin et al, *Nature*, 2017).

Although π -helix has a higher free energy than α -helix, both configurations should be in their local minimum energy to be relatively stable. With this in mind, a channel with a π -to- α activation mode may just need a relatively weak agonist while a channel with an α -to- π activation mode would require an agonist to be relative strong (to bring the channel protein from its low energy, closed state to a high energy, activated state).

•The authors proposed that the selectivity filter of PKD2 acts as a gate, like in TRPV1. If this is in fact the case, the active mutant PKD2 F504P should adopt the open state. However, it appears that the selectivity filter of F504P is in a nonconductive state, very similar to wild-type PKD2. The upper gate (selectivity filter) in TRPV1 clearly acts as an activation gate, based on the comparison between the apo and RTx/DkTx-bound activated states. However, in the PKD2 structure, the selectivity filter adopts non-conducting states. Observing a non-conductive state of the filter could mean the inactive state or simply observed a state that is off the pathway, not necessary suggesting that it will act as a gate. These points should be addressed.

>> Based on the TRPV1 structures in apo state and capsaicin- or RTX/DkTx-bound activated states, agonist capsaicin induced a widening of the lower gate, but little movement of the selectivity filter, whereas RTX/DkTx induced widening of both the lower gate and selectivity filter (see figure below from Cao et al, *Nature*, 2013), suggesting that the lower gate acts as a primary gate for TRPV1 and selectivity filter as a secondary gate involved in RTX/DkTx-induced activation, but not capsaicin-induced activation. In other words, the opening of the lower gate should be sufficient to activate TRPV1 and the selectivity filter may act as a secondary gate, which becomes significant when a hyper-potent agonist such as the combined toxins RTX/DkTx is present.

By comparing structures of PKD2 WT and mutant F604P, we found that the lower gate was widened, while the selectivity filter stays stationary, by the F604P mutation (Fig. 5d and Supplementary Figure 8b in our manuscript), which is similar to the capsaicin-induced TRPV1 activation. Therefore, we agree with the reviewer that these observations in fact suggest that the selectivity filter in PKD2 does not act as a gate in F604P mutation-induced activation. Since PKD2 may possess a number of agonists (at least it is generally believed that PKD2 physiological agonist(s) hasn't been identified), it remains unclear as to whether the selectivity filter could act as a secondary gate in some agonist induced PKD2 activation.

We have now revised the second paragraph in Discussion to compare between the selectivity filter and the lower gate.

•I believe the idea that the pi-helix in the S6 activation gate acts as a hinge, which allows splaying of the helix bundle gate and can also transition to an alpha-helix, was first suggested based on comparison of the TRPV2 and TRPV1 channels (Zubcevic et al, NSMB, 2016). The authors should be transparent and clearly acknowledge the origin of this idea in the text.

>> We are grateful to reviewer for the information and reminding. We have now acknowledged the origin of this idea in the revised manuscript (page 10, lines 1-2; page 12, lines 13-14 from bottom). As well, as mentioned above, we have also incorporated the newly reported α -helix to π -helix transition of TRPV6 (McGoldrick et al, *Nature*, 2018) into our manuscript (page 12, lines 2-5 from bottom).

2. Hydrophobic gating and mutagenesis studies.

The term hydrophobic gating relates mainly to channels that do not possess a classic helix bundle crossing gate, and have a large diameter across the lower pore, which makes them appear to be permeable to ions (i.e. they lack residues that create a physical constriction in the pore). But because of the hydrophobic nature of their pores, the energy barrier for permeation of hydrated ions is so large that their pores are effectively closed. This mechanism, in its pure form, does not seem to apply to the PKD2 channels because the residues that act as a designated gate form a tight constriction within the helix bundle crossing in the PKD

structure. While it is true that the residues forming the helix bundle crossing are hydrophobic, the channels are not only gated by the principle of hydrophobicity, but also by a physical opening of the activation gate (gating – wider or narrower opening). Therefore, the premise for the experimental setting is flawed;

>>We agree with the reviewer that PKD2 channels are not only gated by the gate's hydrophobicity, but also by its physical size.

In fact, all resolved structures of TRP channels including PKD2 in closed states to date revealed that the pore diameters determined by the gate residues are within 5-6 Å (see our revised Supplementary Figure 10), which is comparable to the diameters of hydrated Na⁺ (5.5 Å) and K⁺ (4.6 Å) ions. Therefore, the physical constriction formed by the gate in closed TRP channels may play a role in preventing ion flow, but it alone would be insufficient to completely prevent ion flow, which is probably why gate residues in TRP channels have all evolutionarily been selected to be hydrophobic.

In Fig. 3d and f of our manuscript, replacing PKD2 L677 with smaller/larger hydrophobic residues increased/decreased the channel activity. Data on PKD2L1 also supported the dependence of the channel activity on the gate size in addition to the hydrophobicity (Fig. 2a). Further, our recent study (Zheng et al, *FASEB J*, 2017) on TRPV4, TRPC4 and TRPM8 showed that 1) the basal channel activity is dramatically increased by hydrophilic gate mutations (I715N in TRPV4, I617N in TRPC4 and V976S in TRPM8), and 2) these hydrophilic gate mutants can be further activated by their chemical agonists (GSK1016790A for TRPV4, englerin A for TRPC4 and menthol for TRPM8), suggesting that these TRP channels are indeed gated by both the hydrophobic effect and the physical constriction.

Taken together, the hydrophobic barrier and physical constriction formed by the gate residue would need to work together to tightly block ion permeation in the closed TRP channels. Loss of either one would lead to the channel leaking activity that would be detrimental to cells. In the revised manuscript we have now added the concept of the physical constriction gating and interpreted our mutagenesis data accordingly. We would like to thank the reviewer for raising this essential point.

introduction of hydrophilic residues at the hydrophobic gate will increase conductance if the channel works by the hydrophobic gating mechanism. The problems with the experimental setting are that mutation can influence open probability (gating) as well as ion conductance (permeation). The authors' hydrophobic gating infers that the mutation does not affect gating. This is not the case. The Swartz group performed elegant studies using the Shaker K⁺ channel demonstrating that the mutation of hydrophobic residues at the gate affect the channel gating and their effect on the conductance is minimal (Kitaguchi et al, 2003, Sukhareva M et al, 2003). Along the same line, I believe that the Tucker group introduced a charged residue into the pore-lining S6 and obtained the open structure of KirBac 3.1 (Bavro et al, NSMB 2012). Because of this reason, the authors' interpretation of the PKD2L1 and PKD2 mutagenesis data in the context of purely hydrophobic gating is not convincing.

>> We strongly agree with the reviewer that the hydrophobic theory infers that the hydrophilic substitution at the gate primarily increases the open probability (Po) although it may also affect the single-channel conductance.

Based on the hydrophobic gating theory (see a review from Tucker group entitled "hydrophobic gating in ion channels"), a hydrophobic gate closes the pore through the

so-called liquid-vapor transition of water molecules (see panel (a) in the following figure). A transient liquid state is conducive to water molecules, whereas a vapor state is devoid of water molecules within the hydrophobic pore, which forms an energy barrier to water and ions. Molecular dynamic simulations on model nanopores showed that the hydration rate (probability of the pore being hydrated, which is a counterpart to the P_o in biological channels) is determined by both the pore size and hydrophobicity (see panels b and c).

This hydrophobic gating theory is strongly supported by the Swartz group's study on Shaker K^+ channel (Kitaguchi et al, *J Gen Physiol*, 2003, Sukhareva et al, *J Gen Physiol*, 2003) and our recent study (Zheng et al, *FASEB J*, 2017) which showed that hydrophilic gate mutations substantially increased the P_o in TRPM8 (by single channel measurements) and TRPV4 (by noise analysis), while single-channel conductance remained constant or mildly increased. Our newly acquired data on PKD2 and PKD2L1 using noise analysis (see our response below) also supported this gating mechanism. Therefore, TRP channels, Shaker K^+ and KirBac 3.1 channels, and probably other channels seem to share the hydrophobic gating mechanism.

Figure legend: Principles of hydrophobic gating. (a) Cartoon representation of a cross-section through a model hydrophobic nanopore. Hydrophobic surfaces are shown in yellow, and the membrane is shown in green. In an aqueous solution, these nanopores can switch stochastically between wet and dry states via liquid-vapor transitions within the pore. (b) Dependence of the hydration rate of a hydrophobic nanopore on the pore diameter in the 9-12 Å range. (c) Dependence of the pore hydration rate on the pore diameter and pore hydrophobicity (from Aryal et al, *J Mol Biol*, 2015).

•To maintain their claim of hydrophobic gating, the authors must show that the mutations do not affect the single channel conductance but not gating either by single channel measurement or noise analysis or dose-response curve (if ligand or activating stimuli is known), like the Swartz group did with the Shaker K^+ channel. If authors cannot provide the relevant functional data due to the technical difficulty, authors should determine the structure of PKD2 L677N mutant to test if there is any physical widening of the region at N677 as compared to wild type.

>> Following the reviewer's suggestion, we have performed the noise analysis on PKD2L1 and PKD2 channels according to a previously described method (Gray, Chapter 8, *Microelectrode Techniques, The Plymouth Workshop Handbook, second ed., 1994*). At any given voltage, the mean current $I = NiP_o$, where 'N' is the total number of channels present on the membrane; 'i' is the single channel current amplitude when the channel is opening and P_o is the open probability. Assuming that the channel has only

two states (open and closed), then the current variance, from the binomial theory, is given by: $\text{var}(I) = Ni^2Po(1-Po)$. The mean of the current is given by: $\bar{x} = \frac{1}{N} \sum_{i=1}^N x_i$ and the variance of the current by: $\text{var}(x) = \frac{1}{N} \sum_{i=1}^N (x_i - \bar{x})^2$ where 'N' is the number of points sampled. We estimated the Po and i values (mean \pm SEM) for PKD2L1 and PKD2 channels at -100 mV as follow:

		Po	i
PKD2L1	WT (n=5)	0.12 \pm 0.03	11.2 \pm 1.6 pA
	L557N (n=5)	0.65 \pm 0.1	13.5 \pm 2.2 pA
	A558N (n=5)	0.69 \pm 0.12	13.6 \pm 2.4 pA
PKD2	WT (n=5)	0.007 \pm 0.002	8.2 \pm 1.3 pA
	L677N (n=5)	0.25 \pm 0.06	9.1 \pm 1.5 pA

We found that, for both PKD2L1 and PKD2 channels, hydrophilic gate mutations (L557N and A558N in PKD2L1; L677N in PKD2) substantially increase Po at -100 mV, but the unitary current remained unaffected, which is consistent with our previous single channel data on TRPM8 and noise analysis on TRPV4 (Zheng et al, *FASEB J*, 2017) and Swartz group's study on Shaker K⁺ channel (Kitaguchi et al, *J Gen Physiol*, 2003, Sukhareva et al, *J Gen Physiol*, 2003). These data are in agreement with the hydrophobic gating theory, mentioned above.

•In my view the authors' studies are simply to identify critical residues that form a gate. In PKD2L1, substitutions of the gate residue L557 with a polar residue of an equivalent length would influence the gating (pore open wide) rather than the simple change in hydrophobicity. This is actually evident in the mutagenesis data presented; substitutions of Leu577 with longer polar residues cause an increase in channel activity and substitutions with smaller polar residues (S, C, T) give similar results as smaller hydrophobic residues (A, G). This suggests that the increased channel activity may stem from changes in effective pore diameter rather than from decreasing hydrophobicity of the pore, even without invoking the change in gating. Also, substitution of Leu to bulky Trp results in the nearly complete loss of the currents. Trp is not the most hydrophobic amino acid, but it is the largest amino acid. Notably, the Swartz group also observed the Trp mutation caused the channel to prefer the closed state but had little effect on the conductance.

>> As we explained above, TRP channels are assumed to be gated by both hydrophobic barrier and physical constriction formed by the gate residue. Either decrease in the hydrophobicity of gate residue or increase in the pore size would lead to higher open probability and thus larger channel activity. In PKD2L1, the isosteric polar substitution L557N is supposed to increase the hydrophilicity of the gate without a significant change in the pore size, which is an ideal substitution to test the hydrophobic gating in PKD2L1. The resulted substantial increases in the channel activity (Fig. 1b and c) and open probability (see our noise analysis above) are well consistent with the hydrophobic gating theory. The substitutions of gate residue L557 with smaller hydrophobic residues (A and G) also resulted in increases in channel activity (Fig. 2a), suggesting that the PKD2L1 channel is also gated by physical constriction of the pore gate. When it comes to the substitution of L557 with smaller polar residues (S, C and T), both the

hydrophilicity of the gate and the pore size are increased, which together lead to increases in PKD2L1 channel activity. Compared with residue A or G, the residue S, C or T has a higher hydrophilicity, but has a larger size, which is presumably why substitution of L557 with S, C, or T gave similar increase in PKD2L1 channel activity as substitution of L557 with A or G (Fig. 2a). Also, residue N has a larger size than S, C, T, A or G, but has a higher hydrophilicity, and L557N mutant showed similar channel activity as mutants L557S, L557C, L557T, L557A and L557G (Fig. 2a). Taken together, all these mutagenesis data could be interpreted in the context of combined gating by the gate hydrophobicity and physical size. As mentioned above, we have now introduced physical constriction gating in the manuscript and re-interpreted the mutagenesis data. Based on this new hypothesis, the loss-of-function of L557W could now be interpreted as a result of tighter physical constriction formed by the bulky W residue, which is consistent with Swartz group's data on W mutation introduced to the gate residue of Shaker K⁺ channel.

•It looks like that the PKD construct that was used for functional studies is full-length, not the truncated construct that was used for cryo-EM studies. The authors should provide data showing that the construct that was used for cryo-EM (53-792) is functional and that functional phenotype remains unchanged when mutated (F604P).

>> **Good point. The reason why we used full-length PKD2 protein in our functional study but the truncated E53-D792 protein in our structural study is that this truncated version of PKD2 protein was shown to exhibit enhanced biochemical stability and structural homogeneity over the full-length protein (Shen et al, *Cell*, 2016), making it a better choice for structure determination with cryo-EM. We have now specified this in the revised manuscript (page 9, lines 7-10). Following the reviewer's suggestion, we tested the function of PKD2 E53-D792 truncated mutant. Since the basal activity of WT PKD2 is not detectable we compared the function of full-length PKD2 and truncated version E53-D792 in the presence of the F604P mutation. We found that indeed they exhibit similar functional activities (as Supplementary Figure 3 in the revised manuscript).**

•The authors use the term hydrophobic seal distinctly from physical steric occlusion of the pore without much explanation. This is quite confusing and incorrect. Hydrophobic seal was originally used to describe the hydrophobic core of the voltage-sensor that physically occludes ion permeation through the voltage sensor. Also, the Tucker and Sansome's original paper of hydrophobic gating did not use hydrophobic seal. In several places, I think authors interchange hydrophobic seal and physical steric occlusion. The authors should define something like a hydrophobic barrier, in which the opening is large enough to permeate ions, but due to its hydrophobicity cannot.

>> **We have now defined hydrophobic barrier in the manuscript (page 4, lines 8-9) and changed the term "hydrophobic seal" to "hydrophobic barrier" throughout the manuscript.**

Minor points:

•Pi-helices in S6 have been observed in other WT PKD channels and the closely related TRPML channels. It would be useful to have a comparison of the pores in related channels with gate/restricting residues annotated (i.e. a modified version of figure S5 (S6 in legends)) along with a discussion on the questions posed above.

>> Agreed. We have now included structures of PKD2_{apo}, PKD2_{F604P}, PKD2_{MI} (WT PKD2 bound with multiple calcium ions), PKD2_{SI} (WT PKD2 bound with single calcium ion) and closed and open conformations of both TRPML1 and TRPML3 channels in the revised Supplementary Figure 10. Gate residues and pore diameters determined by the gate residues have been shown. We have also added a related discussion on page 11, second paragraph and page 12, second paragraph.

•p.3, 5th line from bottom: activated mutant should be activating mutant

>> **Changed.**

•p. 4, 6th line from top. The sentence is somewhat unclear; it needs a rewrite.

>> **We have now changed it to simple sentence “The molecular gating mechanisms of PKD2 and PKD2L1 channels have remained elusive”.**

•Order of Fig. S4-S6, along with figure legends, is wrong.

>> **Corrected.**

•Figure S5 (S6 in legends) is not very informative (the pi helix and alpha helix in TRP channels). Align the structures so that the same view is obtained for all channels. It would also be helpful to show gate residues for each of the channels. Also, it'd be nice to see both open and closed conformations for channels where both are available (i.e. TRPV1).

>> **Excellent points. We have now aligned the structures to the same view. The gate residue and the pore diameter determined by the gate residue have been shown for each channel. As suggested, we have included both open and closed conformations for PKD2, TRPML1, TRPML3, TRPV1 and TRPV6. Besides, as the structures of TRPM4 and TRPM8 were published when this manuscript was under review, we have now also included them (as revised Supplementary Figure 10).**

•The in vivo complementation assays utilize L677G, which is not the most potent GoF mutant of the PKD2 channel. It does not appear to be explained in the text why this mutant was preferred over L677N. In addition, complementation with WT PKD2 does not prevent formation of pronephric cysts; is there a reason for this? The in vivo assay does not add much value to the study. I would recommend removing this section from the MS and moving the corresponding figure to supplementary information.

>> **We thank the reviewer for raising the question about complementation assays in zebrafish. In fact, we included L677N mutant in our *in vivo* zebrafish assays, but this mutant showed no rescue effect on zebrafish tail curing or pronephric cysts induced by PKD2 knockdown. We do not know the underlying reason; it is possible that mutant L677N exhibits no constitutive channel activity in zebrafish embryos which is different from the *Xenopus* oocyte model. We have now indicated this in the manuscript (page 9, lines 1-3).**

It currently remains unclear why the human WT PKD2 does not prevent formation of pronephric cysts in zebrafish, but to the best of our knowledge this has been observed in a number of labs. Possible reasons include: 1) human PKD2 may not be able to fully fulfill the function of zebrafish PKD2 (63% amino acid identity); 2) human PKD2 may have a low steady-state expression or altered tissue distribution in zebrafish.

Our zebrafish complementation assays provided a further support for the gain-of-function of PKD2 gate mutant L677G under an *in vivo* condition, which indicates that L677G is an excellent candidate for generation of PKD2 gain-of-function knockin mice to study PKD2-associated cystogenesis. For these reasons, we would like to keep the zebrafish *in vivo* data.

Reviewer #3 (Remarks to the Author):

This study examined the gating of PKD2 and PKD2L1 and suggested a novel hydrophobic gating mechanism which provided insights into how these channel function. A strength of this work is that multiple techniques were employed to examine the gating of these channel proteins. Two-electrode voltage clamp in frog oocytes was used to identify critical residues that would lead to gain-of-function (GOF) or loss-of-function (LOF) mutations in the proteins. They then expressed these GOF mutant proteins to rescue PKD2 knockdown zebrafish, where (the KD fish were sick without the GOF PKD2. They then obtained high resolution structures of PKD2 to untangle the changes in the protein that lead to the altered gating. The authors have developed a clever model that suggests that the closing the pore occurs through forming a hydrophobic seal. A combined twisting and splaying motion of gate residues during PKD2 activation leads to opening of the hydrophobic gates. Moreover, they show that PKD2L1 utilizes two residues for the hydrophobic gate and PKD2 utilizes a single residue.

There are limitations to this study. One limitation is that it is assumed that the reader will know that the full length protein is expressed in cells and fish to get functional results and the truncated protein is utilized to examine the structural changes. This difference needs to be explicitly stated. Also, the limitations of comparing function using a full protein and structure using a truncated protein needs to be discussed.

>> We thank the reviewer for raising this important point, which was in fact also raised by the second reviewer. We have now explicitly stated in the manuscript why a truncated PKD2 protein is utilized for structural determinations (page 9, lines 7-10). We have also described limitations of comparing the function using a full protein and the structure using a truncated protein in Discussion (page 12, lines 9-12 from bottom). In addition, we tested the channel function of the truncated mutant protein E53-D792 and found that it has similar channel function as the full-length PKD2, i.e., both the truncated and full-length PKD2 showed constitutive channel activities in the presence of the F604P mutation. The data have now been included as Supplementary Figure 3 in revised manuscript.

Another limitation is the lack of acknowledgment that calcium is an agonist of PKD2 channel activity and PKD2 dependent currents. The statement (on page 3) the agonist for PKD2 homotetrameric channel has not been identified is misleading. It is true that it would be ideal to identify an agonist other than calcium. However, there are many publications showing that the PKD2 channel is activated by cytoplasmic calcium. Both single channel measurements and calcium release from intracellular stores are activated by increases in cytoplasmic calcium. The effects of calcium as an agonist must be addressed here and throughout the

manuscript. For example, at the bottom of page 4, the authors say that the ligand-induced open state structure cannot be resolved until an agonist is identified. But the effects of calcium are not considered. And at the beginning of the Allosteric section of Results on page 8 it is stated that the agonist of PKD2 is unknown without mentioning calcium as a possible agonist. Of course, the authors may want to ignore calcium as an agonist because the calcium binding sites that may confer ligand-induced openings are in the cytoplasmic domains, regions that are not included in the structural analysis. This is a problem of the study, and it must be addressed in this manuscript.

>> This is an important point and we agreed to be inclusive. We have now acknowledged throughout the manuscript that calcium is an agonist and regulator of PKD2 channels. Specifically, 1) on page 3, we have replaced the statement “the agonist for PKD2 homotetrameric channel has not been identified” with “PKD2 homotetrameric channels were reported to be regulated by cytoplasmic Ca²⁺ in a bell-shape manner, i.e., Ca²⁺ up-regulates the PKD2 open probability at low concentrations while exhibits an inhibition at elevated levels”; 2) on page 4, we have replaced the sentence “However, ligand-induced open state structure is unlikely to be resolved to uncover the PKD2 activation mechanism until an agonist is identified” with “Therefore, a PKD2 structure in an activated state is needed to gain insights into the PKD2 activation mechanism and associated conformational changes”; 3) on page 8, we have deleted the statement “the agonist of PKD2 homotetrameric channel is unknown”; and 4) We have added a paragraph to discuss how calcium as an PKD2 agonist could induce different conformational changes, versus channel opening by the F604P mutation (page 13, second paragraph).

In the introduction, the authors claim that the hypothesis that PKD1 and PKD2 act as a fluid flow sensor has been challenged and they cite the work of only one group. This is not correct as a number of groups have experimental results that question this hypothesis. At the same time, several groups have strong evidence to support the hypothesis. The relevance of cilia to clinical outcomes is also unclear. If these ideas are going to be discussed, a more balanced approach is needed.

>> Agreed. We have now changed it to “PKD1 and PKD2 have been shown to form a complex and proposed to sense fluid flow on the membrane of primary cilia. However, this hypothesis has remained debatable (refs).” and included six citations here in which three were in support of this hypothesis (Xu et al, *Am J Physiol Renal Physiol*, 2007; Nauli et al, *Circulation*, 2008 and Boehlke et al, *Nat Cell Biol*, 2010) while other three challenged it (Ma et al, *Nat Genet*, 2013; Stavola et al, *FASEB J*, 2016 and Delling et al, *Nature*, 2016).

The statement Compared with PKD2, its homologue PKD2L1 (also called polycystin-2L1 or TRPP3), which is not involved in ADPKD, has been more extensively studied as an ion channel is not correct. The authors need to be certain that their statements such as this can be supported unequivocally.

>> We have now changed it to “PKD2L1 (also called polycystin-2L1 or TRPP3) is a homologue of PKD2 and shares 54% amino acid identity with PKD2, but is not involved in ADPKD.”.

On page 5, in the statement Compared to the general functional characteristics of PKD2 and PKD2L1 channels, it has remained more elusive, it is not clear what has remained elusive.

>> It has now been replaced with "The molecular gating mechanisms of PKD2 and PKD2L1 channels have remained elusive."

Bottom of the first paragraph on page 5, is it correct to refer to L577 and A558 or should it be L557 and A558 ? Also in Fig 1 panel B the currents are labeled 577 and 578, but in other panels of Fig 1 it is 557 and 558. This is very confusing. Please check and correct. In fact, please check all residue numbers throughout the manuscript.

>> Corrected across the manuscript.

Bottom of page 7. pronouncing should be pronounced . And observed larger channel activity should be observed larger current – channel activity is not measured, ion currents are measured here.

>> All changed.

The effect shown in Fig 4 D is claimed to be a significant rescue but the effect is only 50%. An explanation of why the effect is not more complete is needed.

>> Excellent point. We have now included the following potential reasons in the revised manuscript (page 8, lines 2-5 from bottom). Human PKD2, the protein used in our electrophysiological studies, was employed in zebrafish rescue experiments, so the only 50% rescue effect in pronephric cysts in Fig. 4D could be because 1) functional differences between human PKD2 and zebrafish PKD2 which shares moderate 63% amino acid identity; 2) the gain-of-function PKD2 mutants identified by *Xenopus* oocyte expression system may show lower channel activity in zebrafish pronephric cells, leading to limited rescue effect.

The size of the PKD2 pore has been functionally assessed previously (Anyatonwu JBC 2005). This work needs to be included in the comparison.

>> We have now cited this work in Discussion (page 13, lines 16–17 from bottom).

On page 11, there is speculation about the different effects that would be evoked by one or two molecules of calcium binding. The effect of the number and location of EF hand motifs in the cytoplasmic domain of PKD2 was studied and found to regulate the calcium dependence of PKD2 function (Kuo, FASEB J 2014). Discussion of how calcium binding in the cytoplasmic domain needs to be included in the manuscript.

>> We have now discussed how calcium may bind to the PKD2 C-terminal EF-hand domain and how this binding may modulate PKD2 gating in the revised manuscript (page 13, second paragraph).

Reviewers' comments:

Reviewer #1 (Remarks to the Author):

To summarise my previous comments, to the best of my knowledge this work is novel and of interest to the community.

This paper definitely provides information that will provoke discussion within the polycystic kidney disease community.

There is one recent paper that perhaps they should refer to, since it does suggest a number of changes in the way that these channels work, so it would be useful if in the discussion they took into account this work: Liu et al., E.Life 2018, <https://elifesciences.org/articles/33183>

I would not say that the authors should change their interpretation of their work to align completely with the work in this paper, there are a number of interesting reinterpretations from the Liu paper which don't necessarily agree with current thinking eg PKD2 is a Na/K channel, regulated by Ca, not a Ca channel, and that PKD1 and PKD2 do not have to form a complex to traffic to the cell membrane. It would be good to have a paragraph comparing the results, assessing the . The Liu paper certainly does not negate the results in this paper.

Line 72: A reference for the Tolvaptan results would be useful, if one is available.

Pg 6, Line 164: "mutants 558T" should be A558T

Pg 10. line329: the phrase "a novel gating mechanism in PKD2" is unclear. Do they mean a novel gating mechanism for one particular channel, or do they mean that this is novel in general? The mechanism is certainly similar to the gating of other TRP and other channels, so it is not novel with respect to other channels, but it is perhaps novel, although not unexpected, for PKD2. This is discussed in the discussion section, so it would be better to remove these comments from the previous paragraph.

In general I feel that the authors have responded effectively to many of the queries raised by this reviewer and others. They have clearly worked hard to address the reviewers comments and I think the work is now acceptable for publication. They have also corrected many of the errors of language, so the paper is now much easier to read.

It is an impressive set of results, definitely worthy of publication in Nat. Comms. in my opinion.

Reviewer #2 (Remarks to the Author):

The authors improved their revision significantly by performing additional experiments and providing structural analyses. This paper provides an important insight into the mechanism of channel gating by identifying gate-forming residues in PKD2 and showing structural changes of PKD2. However, I disagree with the authors' claim that the newly included functional (Po) data support the hydrophobic gating theory, and I recommend that authors revise their interpretation.

Major comments

On page 5,

"We also performed noise analysis on whole-cell current traces

146 obtained at -100 mV and found that both mutants L557N and A558N have much higher open

147 probability than WT PKD2L1 (0.65 ± 0.10 for mutant L557N, 0.69 ± 0.12 for mutant A558N
148 and 0.12 ± 0.03 for WT PKD2L1, $n = 5$), whereas their single-channel currents were
comparable

149 (13.5 ± 2.2 pA for mutant L557N, 13.6 ± 2.4 pA for mutant A558N and 11.2 ± 1.6 pA for WT
150 PKD2L1, $n = 5$), which is in agreement with the hydrophobic gating theory 28 and consistent
151 with previous studies on Shaker K⁺ channels 29,30.”

Mutations of the gate-forming residues in the channels with the helical-bundle-crossing (HBC)
mechanism will alter P_o , which cannot be interpreted as supporting evidence for hydrophobic
gating theory. Rather the new results indeed indicate that PKD2 operates by HBC mechanism.

Therefore authors should revise the underlined part to something like “which agrees well with the
helical bundle crossing (physical constraint) model, suggesting it is difficult to exclude the
possibility that helical bundle crossing model plays a role in PKD2 gating”.

As authors noted, the essence of the hydrophobic gating is that pore can adopt a closed state even
where the size of gate is big enough to allow ion permeation because of the hydrophobic residues
provides an energetic barrier. This model contrasts the classical helical-bundle-crossing gating
(HBC) mechanism where the pore adopts the closed state by physically occluding the ion
permeation (a physical barrier).

Thus, the Tucker and Sansome groups did MD simulation studies of TWIK-1 and found that the
mutation of L146 to either a polar (L146N) or an acidic residues (L146D) altered the
hydrophobicity in the ion permeation pathway, but not the closed state structure (Nat. Commun,
2014). They subsequently found the mutant channels increased activity, which was interpreted as
mutations rendered the closed state of TWIK-1 “leaky (altered conductance)”, not necessarily
destabilizing closed state “gating (altered P_o)”. Mutation of the gate-forming residues according to
the hydrophobic-gating model should have minimal influence on gating because these residues are
substantially far apart. On the contrary, based on the helical-bundle-crossing (HBC or physical
constriction) gating model, the gate-forming residues, when mutated, have significant effects on
gating, either by stabilizing an open state or by decreasing a closed state, which result in an
increase in P_o . This is clearly demonstrated by the Swartz group (reference 29 and 30 in the text).
Interestingly, both Tucker group and this paper used the Swartz group’s studies to support the
hydrophobic gating model, which is incorrect because the conclusion of the Swartz’s papers is in
line with the HBC model.

This paper nicely demonstrates the gate-forming residues, and showed structural changes that is
associated with gating. However, the current data in this manuscript does not support the
hydrophobic gating model, and I am not sure why authors would like to maintain their view with
the hydrophobic gating model. Hydrophobicity in the gate or the cavity regions likely play an
important role in channel gating, but whether the importance of hydrophobicity in these regions is
through the model put forth by Tucker and Sansom groups remain to be seen.

Reviewer #3 (Remarks to the Author):

The revised version of this manuscript is improved.

The authors now state the full length protein is expressed in cells and fish to get functional results
and the truncated protein is utilized to examine the structural changes. But the limitations of
comparing function using a full protein and structure using a truncated protein are not explained.
This part of the manuscript still needs to be expanded.

There still are language issues, awkward sentences and grammar errors.

Responses to reviewers' comments (bold and start with ">>"; others are referees' original comments)

Reviewer #1 (Remarks to the Author):

To summarise my previous comments, to the best of my knowledge this work is novel and of interest to the community. This paper definitely provides information that will provoke discussion within the polycystic kidney disease community.

>> We are grateful to the reviewer for the very positive view of our work.

There is one recent paper that perhaps they should refer to, since it does suggest a number of changes in the way that these channels work, so it would be useful if in the discussion they took into account this work: Liu et al., E.Life 2018, <https://elifesciences.org/articles/33183>

I would not say that the authors should change their interpretation of their work to align completely with the work in this paper, there are a number of interesting reinterpretations from the Liu paper which don't necessarily agree with current thinking eg PKD2 is a Na/K channel, regulated by Ca, not a Ca channel, and that PKD1 and PKD2 do not have to form a complex to traffic to the cell membrane. It would be good to have a paragraph comparing the results, assessing the. The Liu paper certainly does not negate the results in this paper.

>> Excellent suggestion. We have now added a paragraph in Discussion (page 13 and 14) to discuss the results from Liu paper.

Line 72: A reference for the Tolvaptan results would be useful, if one is available.

>> A reference for the Tolvaptan results has now been included.

Pg 6, Line 164: "mutants 558T" should be A558T

>> Corrected.

Pg 10. line329: the phrase "a novel gating mechanism in PKD2" is unclear. Do they mean a novel gating mechanism for one particular channel, or do they mean that this is novel in general? The mechanism is certainly similar to the gating of other TRP and other channels, so it is not novel with respect to other channels, but it is perhaps novel, although not unexpected, for PKD2. This is discussed in the discussion section, so it would be better to remove these comments from the previous paragraph.

>> We mean that it is a novel gating mechanism for PKD2. Following the reviewer's suggestion, we have now deleted these comments in sections (page 4, line 10 from bottom; page 10, line 5 from bottom) before the Discussion section.

In general I feel that the authors have responded effectively to many of the queries raised by this reviewer and others. They have clearly worked hard to address the reviewers comments and I think the work is now acceptable for publication. They have also corrected many of the errors of language, so the paper is now much easier to read.

It is an impressive set of results, definitely worthy of publication in Nat. Comms. in my opinion.

Reviewer #2 (Remarks to the Author):

The authors improved their revision significantly by performing additional experiments and providing structural analyses. This paper provides an important insight into the mechanism of channel gating by identifying gate-forming residues in PKD2 and showing structural changes of PKD2. However, I disagree with the authors' claim that the newly included functional (P_o) data support the hydrophobic gating theory, and I recommend that authors revise their interpretation.

>> We are very thankful to the reviewer for the detailed explanations on the hydrophobic gating and helical-bundle-crossing gating.

Major comments

On page 5,

“We also performed noise analysis on whole-cell current traces

146 obtained at -100 mV and found that both mutants L557N and A558N have much higher open

147 probability than WT PKD2L1 (0.65 ± 0.10 for mutant L557N, 0.69 ± 0.12 for mutant A558N

148 and 0.12 ± 0.03 for WT PKD2L1, $n = 5$), whereas their single-channel currents were comparable

149 (13.5 ± 2.2 pA for mutant L557N, 13.6 ± 2.4 pA for mutant A558N and 11.2 ± 1.6 pA for WT

150 PKD2L1, $n = 5$), which is in agreement with the hydrophobic gating theory 28 and consistent

151 with previous studies on Shaker K^+ channels 29,30.”

Mutations of the gate-forming residues in the channels with the helical-bundle-crossing (HBC) mechanism will alter P_o , which cannot be interpreted as supporting evidence for hydrophobic gating theory. Rather the new results indeed indicate that PKD2 operates by HBC mechanism.

>> We agree with the reviewer that "altered P_o cannot be used as supporting evidence for hydrophobic gating theory". This is because both the hydrophobic gating theory (*J Mol Biol* 2015 by Tucker and Sansome groups) and HBC model predict increased P_o due to hydrophilic mutations at gate site (see our detailed explanations below).

Therefore, the two models do not contradict with each other in terms of P_o (also see our responses below regarding Tucker and Sansome's study in *Nat Commun* 2014).

Channels that fit with the hydrophobic gating model can increase the P_o and single-channel conductance with unaffected gating by gate hydrophilic mutations, e.g., MscL

channel (Birkner et al, *PNAS*, 2012) and MthK channel (Shi et al, *J Mol Biol*, 2011). In our recent publication, we found in TRPM8 and TRPV4 that, in addition to increased single-channel currents, P_o is also dramatically increased by gate hydrophilic substitutions (Zheng et al, *FASEB J*, 2018).

Therefore authors should revise the underlined part to something like “which agrees well with the helical bundle crossing (physical constraint) model, suggesting it is difficult to exclude the possibility that helical bundle crossing model plays a role in PKD2 gating”.

As authors noted, the essence of the hydrophobic gating is that pore can adopt a closed state even where the size of gate is big enough to allow ion permeation because of the hydrophobic residues provides an energetic barrier.

This model contrasts the classical helical-bundle-crossing gating (HBC) mechanism where the pore adopts the closed state by physically occluding the ion permeation (a physical barrier).

Thus, the Tucker and Sansome groups did MD simulation studies of TWIK-1 and found that the mutation of L146 to either a polar (L146N) or an acidic residues (L146D) altered the hydrophobicity in the ion permeation pathway, but not the closed state structure (Nat. Commun, 2014). They subsequently found the mutant channels increased activity, which was interpreted as mutations rendered the closed state of TWIK-1 “leaky (altered conductance)”, not necessarily destabilizing closed state “gating (altered P_o)”.

>> In this study by the same Tucker and Sansome groups, the channel activity is assessed by either whole-cell or giant patch recordings, both of which did not provide single-channel parameters. Thus, increased channel activity means increased whole-cell current (or whole-cell conductance), i.e., $N \cdot P_o \cdot i$ is increased (where "i" stands for the single-channel current) but it remained undetermined as to which of P_o or i , or both, is increased. However, in their subsequent *J Mol Biol* 2015 paper, they stated "both sequence and structural alignments suggest that the hydrophobic cuff in TWIK-1 is equivalent to the hydrophobic constriction formed by residue Ala88 in MthK". By this, they would suggest that TWIK-1, like MthK, have both increased P_o and single-channel conductance by gate hydrophilic substitutions.

Mutation of the gate-forming residues according to the hydrophobic-gating model should have minimal influence on gating because these residues are substantially far apart. On the contrary, based on the helical-bundle-crossing (HBC or physical constriction) gating model, the gate-forming residues, when mutated, have significant effects on gating, either by stabilizing an open state or by decreasing a closed state, which result in an increase in P_o . This is clearly demonstrated by the Swartz group (reference 29 and 30 in the text). Interestingly, both Tucker group and this paper used the Swartz group's studies to support the hydrophobic gating model, which is incorrect because the conclusion of the Swartz's papers is in line with the HBC model.

>> We agree with the reviewer that in channels that fit with the HBC (physical constriction) gating model, mutations in the gate-forming residues have significant effects on gating, either by stabilizing an open state or by decreasing a closed state, which results in an increase in P_o . In this case, gating is equivalent to P_o . We also agree

with the reviewer that in channels that fit with the hydrophobic gating model, mutations in the gate-forming residues have minimal influence on gating because these residues are substantially far apart. However, in this case, hydrophilic gate mutations can still dramatically increase P_o , presumably through disrupting the hydrophobic barrier according to the hydrophobic gating theory. This is demonstrated in the hydrophobic gating channels MscL (Birkner et al, *PNAS*, 2012) and MthK channel (Shi et al, *J Mol Biol*, 2011). In our recent publication (Zheng et al, *FASEB J*, 2018) our single-channel patching and noise analysis data on TRPM8 and TRPV4 also supported this concept. Therefore, in channels that fit with hydrophobic gating model, gating can be very different from P_o , i.e., mutations in the gate residue can alter P_o while gating remains unaffected. Therefore, altered P_o by hydrophilic gate mutations cannot be used to distinguish between the HBC and hydrophobic gating models.

This paper nicely demonstrates the gate-forming residues, and showed structural changes that is associated with gating. However, the current data in this manuscript does not support the hydrophobic gating model, and I am not sure why authors would like to maintain their view with the hydrophobic gating model. Hydrophobicity in the gate or the cavity regions likely play an important role in channel gating, but whether the importance of hydrophobicity in these regions is through the model put forth by Tucker and Sansom groups remain to be seen.

>> Based on the explanations provided by the reviewer and us above regarding the two models, we agree to be inclusive and to soften our related statement. For this, we think that it would be much better to change the original sentence ", which is in agreement with the hydrophobic gating theory 28 and consistent with previous studies on Shaker K^+ channels 29,30" to ". The increased open probability by gate hydrophilic substitutions is consistent with the hydrophobic gating theory (28), although the helical-bundle-crossing (or physical constriction) mechanism may also play a role in the PKD2L1 gating". We truly appreciate the reviewer's help and efforts on this important point and really hope that the reviewer will accept the change.

Reviewer #3 (Remarks to the Author):

The revised version of this manuscript is improved.

The authors now state the full length protein is expressed in cells and fish to get functional results and the truncated protein is utilized to examine the structural changes. But the limitations of comparing function using a full protein and structure using a truncated protein are not explained. This part of the manuscript still needs to be expanded.

>> We have described limitations of comparing the function using the full-length PKD2 protein and the structure using a truncated PKD2 protein in Discussion (page 12, lines 10-13 from bottom).

There still are language issues, awkward sentences and grammar errors.

>> The language has now been carefully polished throughout the manuscript by several coauthors.

REVIEWERS' COMMENTS:

Reviewer #2 (Remarks to the Author):

Authors responded well to my criticism. However, their suggested change of the original sentence is not appropriate; "The increased open probability by gate hydrophilic substitutions is consistent with the hydrophobic gating theory (28), although the helical-bundle-crossing (or physical constriction) mechanism may also play a role in the PKD2L1 gating". Because mutations did not affect single channel conductance but only open probability, the effect of the mutations is more in line with the helical-bundle-crossing mechanism.

I suggest changing the sentence to "The increased open probability without significant changes in single channel conductance by gate hydrophilic substitutions suggests that the helical-bundle-crossing (or physical constriction) mechanism play a significant role in PKD2L1 gating in addition to the hydrophobic gating mechanism".

Otherwise, the manuscript is in a good shape. Again, this paper provides important insights into PKD2 gating, which can be extended to other TRP channel proteins.

Responses to reviewers' comments (bold and start with ">>"; others are referees' original comments)

Reviewer #2:

Authors responded well to my criticism. However, their suggested change of the original sentence is not appropriate; "The increased open probability by gate hydrophilic substitutions is consistent with the hydrophobic gating theory (28), although the helical-bundle-crossing (or physical constriction) mechanism may also play a role in the PKD2L1 gating". Because mutations did not affect single channel conductance but only open probability, the effect of the mutations is more in line with the helical-bundle-crossing mechanism.

I suggest changing the sentence to "The increased open probability without significant changes in single channel conductance by gate hydrophilic substitutions suggests that the helical-bundle-crossing (or physical constriction) mechanism play a significant role in PKD2L1 gating in addition to the hydrophobic gating mechanism".

Otherwise, the manuscript is in a good shape. Again, this paper provides important insights into PKD2 gating, which can be extended to other TRP channel protein

>> As the reviewer suggested, we have now changed the sentence to "The increased open probability without significant changes in single-channel current suggests that the helical-bundle-crossing (or physical constriction) mechanism plays a significant role in PKD2L1 gating in addition to the hydrophobic gating mechanism."